# FairLoRA: Targeted Bias Mitigation without Performance Loss

## Abstract

Ensuring fairness in machine learning models is critical, but existing debiasing techniques often sacrifice model performance, struggle to adapt to emerging biases, or require extensive sensitive attribute annotations. To address these challenges, we propose FairLoRA, a novel low-rank adaptation method that mitigates bias while preserving model performance. FairLoRA incorporates parameter-efficient modular LoRA components, enabling iterative bias mitigation to ensure fairness across multiple sensitive attributes without interfering with previous adjustments. Furthermore, it employs discriminators to identify biased classes with reduced reliance on sensitive information, significantly reducing the need for annotated data. We theoretically derive conditions under which FairLoRA fine-tuning can effectively mitigate bias while maintaining the original model's performance. We then empirically validate its effectiveness across diverse computer vision and natural language processing tasks. Our experimental results show that, even for models that have undergone prior bias mitigation training, the integration of FairLoRA fine-tuning can further enhance fairness, while maintaining or even slightly improving the original performance.

## 1 Introduction

Machine learning (ML) models demonstrate immense power and achieve remarkable success in both computer vision (CV) and natural language processing (NLP) domains. As ML models have been widely applied to many critical fields in our society, fairness concerns have recently gained increasing attention in their research and applications (Liu et al., 2023). For example, Gong et al. (2021) observe that applying biased face recognition systems can cause potential risk in law enforcement. Lu et al. (2024) highlight that transformers-based models make biased predictions in CV and NLP fields. Therefore, algorithmic fairness is a burgeoning topic of broad interest, and addressing fairness issues in ML models is a significant but challenging task.

The main cause of fairness issues is that ML methods have provided opportunity for negative societal biases to affect the models through data. Traditional definitions of algorithmic fairness often focus on performance disparities among different demographic groups. The standard approach of empirical risk minimization (ERM) trains ML models to minimize average loss on a training set. However, ERM method can produce models that achieve high accuracy on average but still consistently fail on rare and atypical groups of examples (Song et al., 2024). These kinds of performance disparities across groups can be especially pronounced in the presence of data that encode negative societal biases (Ferrara, 2023) and other spurious correlations (Neuhaus et al., 2023): misleading heuristics that work for most training examples but do not always hold (Sagawa et al., 2020a). For example, in the task of toxic comment classification, the training data is often biased by correlating toxicity with particular demographic identities (e.g., certain races or religions) (Mathew et al., 2021). Therefore, models that learn this spurious correlation will reflect the biases in these datasets, and cause fairness issues in many applications, such as language tasks (McCoy et al., 2019), facial recognition (Sagawa et al., 2020a), and medical imaging (Oakden-Rayner et al., 2020).

Existing works that attempt to address fairness issues in ML can be broadly classified into two categories: model interventions and data interventions (Jain et al., 2024). Model interventions target either model weights (Santurkar et al., 2021; Shah et al., 2024) or the training procedure (Sagawa et al., 2020a; Kirichenko et al., 2023). However, most of previous works need to fine-tune all

the parameters in the bias-mitigation process, and can not maintain the model performance while improving the fairness of some demographic group. Therefore, it is rather hard to combine different debiasing methods together to fully utilize their advantages, and it is also difficult to mitigate bias of different sensitive attributes as improving the fairness of one demographic may affect another. Moreover, previous works still suffer from challenges in trade-off between accuracy and fairness, high-demand computational resources, and expensive annotations for sensitive information:

(1) Most previous works struggle to enhance the fairness of ML models while maintaining performance, and often focus solely on a single sensitive attribute (Liu et al., 2023). Furthermore, many existing approaches lack theoretical guarantees for the trade-off between fairness and performance, leaving a significant gap in our understanding of these critical relationships.

(2) When addressing fairness issues in large-scale pre-trained models, numerous existing methods necessitate fine-tuning all parameters to achieve a balance between fairness and accuracy. However, updating such a vast number of parameters can be prohibitively expensive (Petersen et al., 2021) and may lead to catastrophic forgetting, potentially diminishing the model's efficacy in other tasks.

(3) Traditionally, previous approaches have operated either in a full-information setting, where group labels are required for each training example, or in a no-information setting, where all group labels are unavailable (Liu et al., 2021). While full-information methods empirically demonstrate superior performance compared to no-information approaches, obtaining training group annotations is often costly and time-consuming. Consequently, there remains a crucial need for further exploration of partial-information settings which utilize only a small portion of group labels.

To solve the above challenges, we propose **FairLoRA**, a novel fine-tuning method to enhance fairness of ML models without degrading model performance. By combining a group discriminator with a low-rank adaptation (LoRA) block trained on group-balanced subset of the data, the FairLoRA block can reduce the worst-group error (Sagawa et al., 2020a) and thus improve the fairness of ML models. The main contributions of our work are summarized as:

(1) FairLoRA fine-tuning can enhance model fairness while maintaining its performance, supported by theoretical analysis that provides guarantees. FairLoRA module offers high flexibility, allowing it to be combined with other debiasing methods for further fairness improvements. Moreover, following an iterative residual learning paradigm, FairLoRA can address fairness concerns across multiple sensitive attributes.

(2) FairLoRA leverages the representational power of the base model in the group discriminator and the efficiency of the LoRA method, resulting in significantly lower computational costs compared to full-parameter fine-tuning. The group discriminator functions as a gate unit, determining the activation of the LoRA block, which can effectively mitigate catastrophic forgetting issues.

(3) FairLoRA operates under a partial-information setting, where group labels are observed only for a subset of the training set. This approach substantially reduces annotation costs for sensitive attributes compared to full-information settings, making it more practical for real-world applications.

## 2  RELATED WORK

We review fair machine learning work on the trade-off between fairness and performance, fairness-aware finetuning methods, and fairness with/without demographics information.

### 2.1  TRADE-OFF BETWEEN FAIRNESS AND PERFORMANCE

Traditional bias mitigation techniques often involve data preprocessing methods such as re-sampling, re-weighting, or data augmentation to balance datasets across sensitive attributes (Calmon et al., 2017; Liu et al., 2023). While effective to some extent, these methods may not address biases inherent in model architectures or training procedures. To mitigate model-level biases, fairness constraints and regularization terms have been integrated directly into training objectives. Agarwal et al. (2018) proposed a reduction approach transforming fairness-constrained classification into a sequence of cost-sensitive classification problems. Recent works have focused on improving group fairness via distributionally robust optimization. Sagawa et al. (2020a) presented GroupDRO, mini-

mizing the worst-case loss over predefined groups to enhance fairness. However, such methods can increase computational complexity and may negatively impact overall performance.

Balancing fairness and performance remains a critical challenge. Enhancing fairness often results in decreased accuracy, particularly for the majority group (Song et al., 2024). Multi-objective optimization frameworks have been proposed to navigate this trade-off. Martinez et al. (2020) presented a minimax Pareto fairness approach to optimize for both fairness and accuracy. Cotter et al. (2019) developed methods for optimizing non-differentiable fairness metrics alongside standard loss functions. Adaptive methods that adjust training strategies based on subgroup performance have also been explored (Hashimoto et al., 2018). Donini et al. (2018) introduced a duality-based approach to enforce fairness constraints without significantly compromising performance. However, these methods may increase computational complexity or require careful hyperparameter tuning. Therefore, it remains an open question to improve the model fairness while maintaining its performance, and theoretical guarantees for the trade-off between fairness and performance still need to be derived.

## 2.2 FAIRNESS-AWARE FINE-TUNING METHODS AND CATASTROPHIC FORGETTING

Fine-tuning pre-trained models is a common strategy for adapting models to specific tasks. However, standard fine-tuning may inadvertently introduce or amplify biases present in pre-trained models (Zhao et al., 2019). Parameter-efficient fine-tuning (PEFT) techniques can significantly reduce the number of trainable parameters. One of the most popular PEFT techniques is Low-Rank Adaptation (LoRA) (Hu et al., 2022), which reduces the training cost by injecting trainable low-rank matrices into each layer. While LoRA improves fine-tuning efficiency, its application to fairness enhancement has been limited. Das et al. (2024) found that directly using low-rank fine-tuning inadvertently preserves undesirable biases and toxic behaviors. Moreover, directly using LoRA fine-tuning may worsen fairness across subgroups and appear less fair via worst subgroup accuracy (Ding et al., 2024). Therefore, PEFT techniques for fairness still need further research.

To deal with multiple sensitive attributes, continual learning framework can be adopted to improve fairness for different demographics step-by-step. Therefore, another challenge for fairness improving method is catastrophic forgetting, the loss of previously learned knowledge during finetuning, which poses a challenge in bias mitigation and continual learning (Zhang et al., 2023). Finetuning for fairness may degrade original task performance, which is undesirable in practical applications. Continual learning techniques mitigate catastrophic forgetting by preserving important parameters. Kirkpatrick et al. (2017) introduced Elastic Weight Consolidation (EWC), adding regularization to prevent significant updates to critical weights. Sun et al. (2020) proposed LAMOL, a method for language modeling that mitigates forgetting through data replay. However, research that integrates such methods into fairness-aware fine-tuning scenarios remains limited and needs further exploration to improve fairness for different demographics.

## 2.3 FAIRNESS AND DEMOGRAPHIC INFORMATION

Most of existing bias mitigation methods leverage demographic information during training to deal with spurious correlations. For example, Sagawa et al. (2020b) reweight or subsample the majority and minority groups; Goel et al. (2021) synthetically expand the minority groups via generative modeling; Zhang et al. (2021) minimize the worst-group loss during training. Although these bias mitigation methods substantially reduce worst-group error, obtaining corresponding group annotations can be extremely expensive. Some previous works consider the no-information setting where all the group labels are unavailable (Liu et al., 2021). However, methods in no-information setting empirically can not perform as well as full-information setting. Instead, we focus on the partial-information setting, leveraging partial group information during training to achieve more consistent bias mitigation while reducing reliance on full group annotations.

## 3 METHODOLOGY

In this section, we introduce FairLoRA, a PEFT approach designed to enhance fairness in machine learning models without requiring comprehensive group annotations during training. As illustrated in Figure 1, FairLoRA harnesses the representational power of pre-trained models, integrating group

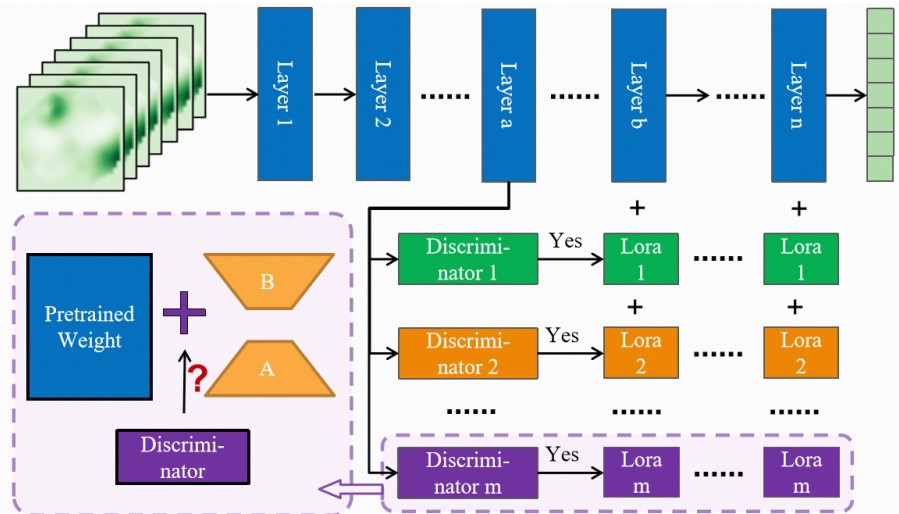

Figure 1: The architecture of FairLoRA, showcasing the integration of discriminators and LoRA blocks across multiple layers. The discriminator is trained beforehand plays a crucial role in determining whether to activate the LoRA block. When the discriminator identifies a sample as belonging to an underrepresented group, the corresponding LoRA block is engaged. Otherwise, the data sample is processed directly by the base model without LoRA intervention.

discriminators and LoRAs to mitigate bias by selectively improving the performance of underrepresented groups.

### 3.1 SENSITIVE ATTRIBUTES AND FAIRNESS DEFINITION

Let $s \in \{0, 1\}$ be the binary sensitive attribute, where $s = 0$ represents the majority group and $s = 1$ represents the minority group. The base model $f_\theta : \mathcal{X} \to \mathcal{Y}$, parameterized by $\theta$, is trained using ERM, minimizing the overall loss $R(\theta) = \mathbb{E}_{(x,y) \sim P}[\ell(f_\theta(x), y)]$, where $\ell(\cdot)$ denotes the loss function, and $P$ represents the data distribution. However, due to data imbalance, the base model tends to perform better on the majority group while underperforming on the minority group. When there are multiple sensitive attributes, we can simply generalize this fairness definition by considering each sensitive attribute in the similar manner.

### 3.2 FAIRLORA STRUCTURE

To mitigate unfairness issues, we propose FairLoRA, a framework consisting of two key components: a group discriminator and a series of LoRA modules. The group discriminator is responsible for identifying whether the input exhibits biases and determining whether corresponding adjustments are necessary. The LoRA modules address the identified biases by making targeted modifications to the model's representations in a low-rank space.

The group discriminator, $D_\phi : \mathcal{X} \to 0, 1$, uses Attention Pooling to aggregate token-level hidden states $h_\theta(x) \in \mathbb{R}^{T \times d}$ from the base model, where $T$ is the sequence length and $d$ is the hidden state dimensionality. The attention pooling mechanism is formulated as $h_{\text{pool}}(x) = \sum_{t=1}^{T} \alpha_t h_\theta(x)_t$, where $\alpha_t = \text{softmax}(w^\top h_\theta(x)_t)$ and $w \in \mathbb{R}^d$ is a learnable weight vector. This pooling results in a global representation $h_{\text{pool}}(x)$, which is considered as a representation of the input sample and used as input to the group discriminator.

LoRA is applied to improve performance for minority groups. We utilize a dataset balanced according to the predicted labels of sensitive attribute categories to train the LoRA modules. In cases where sensitive attribute labels are unavailable, pseudo-labels can be generated using the pre-trained group discriminator. Alternatively, we can also customize the dataset based on task requirements or use other fine-tuning methods to improve fairness.

During inference, the discriminator output $D_\phi(h_{\text{pool}}(x))$ determines whether the LoRA block is activated. The final model's weights are updated as follows:

$$W_{\text{FairLoRA}} = W_{\text{frozen}} + \mathbb{I}(D_\phi(h_{\text{pool}}(x)) = 1) \cdot (BA) \tag{1}$$

where $W_{\text{frozen}}$ is the frozen pre-trained weight matrix of the base model, $B \in \mathbb{R}^{d \times r}$ and $A \in \mathbb{R}^{r \times k}$ are the low-rank matrices introduced by LoRA, with $r \ll \min(d, k)$, and $\mathbb{I}(\cdot)$ is the indicator function, which outputs 1 if $D_\phi(h_{\text{pool}}(x)) = 1$ (minority group), and 0 otherwise. Thus, when $D_\phi(h_{\text{pool}}(x)) = 1$, LoRA is activated to adjust the base model's weights to mitigate bias. Otherwise, data samples are processed directly by the base model without LoRA intervention.

### 3.3 FairLoRA for Multiple Sensitive Attributes

FairLoRA can be extended to handle multiple sensitive attributes $\{s_1, s_2, \ldots, s_k\}$. As shown in Figure 1, for each sensitive attribute $s_i$, a separate LoRA module is introduced. The overall model update after processing all sensitive attributes is formulated as:

$$W_{\text{FairLoRA}} = W_{\text{frozen}} + \sum_{i=1}^{k} \mathbb{I}(D_{\phi_i}(h_{\text{pool}}(x)) = 1) \cdot (B_i A_i) \tag{2}$$

where $B_i \in \mathbb{R}^{d \times r}$, $A_i \in \mathbb{R}^{r \times k}$ are the low-rank matrices corresponding to the $i$-th sensitive attribute, and $D_{\phi_i}$ is the discriminator for attribute $s_i$.

### 3.4 Optimization Framework

The optimization process for FairLoRA involves the following steps:

1. **Group Discriminator Training**: Train the group discriminator to identify the sensitive attribute, using attention pooling to aggregate token-level hidden states for a more accurate representation of the input.

2. **LoRA Fine-Tuning**: Apply LoRA fine-tuning to a dataset balanced according to the predicted labels of the sensitive attribute category to enhance the performance of underrepresented or biased categories, updating $B$ and $A$.

3. **Extend the Chain**: For multiple sensitive attributes, iteratively apply FairLoRA for each attribute, forming a chain of low-rank adaptations.

This approach enables iterative bias mitigation without compromising previous adjustments, ensuring fairness across multiple sensitive attributes while maintaining model performance.

## 4 Theoretical Analysis

In this section, we present a theoretical analysis of FairLoRA, stating key theorems on fairness improvements and performance preservation, along with detailed proofs.

### 4.1 Definitions and Performance Metrics

We begin by defining the key variables and performance metrics used in the analysis.

#### 4.1.1 Group Samples and Model Performance

We define the key variables as follows: $N = N_1 + N_2$, where $N_1$ and $N_2$ are the number of samples in the majority (G1) and minority (G2) groups, respectively. The proportion of minority group samples is $p = \frac{N_2}{N}$.

The model's performance on G1 and G2 is $P(M, \text{G1})$ and $P(M, \text{G2})$. The overall performance is:

$$P(M) = (1 - p) \cdot P(M, \text{G1}) + p \cdot P(M, \text{G2}) \tag{3}$$

Similarly, for the LoRA fine-tuned model:

$$P(M_{\text{LoRA}}) = (1 - p) \cdot P(M_{\text{LoRA}}, \text{G1}) + p \cdot P(M_{\text{LoRA}}, \text{G2}) \tag{4}$$

### 4.1.2 DISCRIMINATOR METRICS

Define the True Positive (TP), False Positive (FP), True Negative (TN), and False Negative (FN) as follows: TP: Correctly classified G2 samples. FP: G1 samples incorrectly classified as G2. TN: Correctly classified G1 samples. FN: G2 samples incorrectly classified as G1.

The True Positive Rate (TPR) and False Positive Rate (FPR) are: $\text{TPR} = \frac{\text{TP}}{N_2}, \text{FPR} = \frac{\text{FP}}{N_1}$

## 4.2 THEORETICAL PROPERTIES OF FAIRLORA

The theoretical results of FairLoRA performance for the majority group and minority group are provided in Lemmas 1 and 2, respectively. The performance preservation condition for the FairLoRA approach is provided in Theorem 1.

**Lemma 1.** For the majority group (G1), the model performance after FairLoRA fine-tuning is:

$$P(M_{\text{FairLoRA}}, \text{G1}) = (1 - \text{FPR}) \cdot P(M, \text{G1}) + \text{FPR} \cdot P(M_{\text{LoRA}}, \text{G1}) \qquad (5)$$

**Lemma 2.** For the minority group (G2), the model performance after FairLoRA fine-tuning is:

$$P(M_{\text{FairLoRA}}, \text{G2}) = \text{TPR} \cdot P(M_{\text{LoRA}}, \text{G2}) + (1 - \text{TPR}) \cdot P(M, \text{G2}) \qquad (6)$$

**Theorem 1.** To ensure that FairLoRA maintains the overall performance of the model, the discriminator's TPR to FPR ratio is required to meet the following condition:

$$\frac{\text{TPR}}{\text{FPR}} \geq \frac{(1 - p)}{p} \cdot \frac{P(M, \text{G1}) - P(M_{\text{LoRA}}, \text{G1})}{P(M_{\text{LoRA}}, \text{G2}) - P(M, \text{G2})} \qquad (7)$$

The proofs for Lemmas 1, 2, and Theorem 1 are provided in the Appendix A.

In summary, our theoretical analysis demonstrates that FairLoRA fine-tuning can effectively mitigate bias while preserving overall model performance by maintaining a suitable TPR-to-FPR ratio. This condition is often achievable in practice, as LoRA fine-tuning aims to improve minority group performance $P(M_{\text{LoRA}}, G2) - P(M, G2)$ while minimally impacting majority group performance $P(M, G1) - P(M_{\text{LoRA}}, G1)$. As a result, the improvement for the minority group typically outweighs the minor effect on the majority group, leading to a manageable threshold for the ratio, which can often be approximated as $\frac{\text{TPR}}{\text{FPR}} \geq \frac{(1-p)}{p}$. Adjusting classification thresholds can also help achieve a high TPR and low FPR, thereby meeting this condition.

When the condition is met, FairLoRA ensures that gains for the sensitive group outweigh losses for the non-sensitive group, enhancing fairness without compromising overall performance.

## 5 EXPERIMENTS

This section presents a comprehensive evaluation of our proposed method, FairLoRA, designed to mitigate biases in pre-trained models while maintaining or improving original performance. We conduct experiments on three widely-used fairness benchmark datasets: CelebA (Liu et al., 2015), MultiNLI (Williams et al., 2018), and HateXplain (Mathew et al., 2021), and evaluate three key scenarios: (1) eliminating a single type of bias, (2) sequentially eliminating multiple types of biases, and (3) evaluating the impact of dataset proportions on FairLoRA. Comparisons with prevalent methods demonstrate that FairLoRA consistently improves fairness metrics without significant performance loss, and in some cases, even enhances overall accuracy.

## 5.1 EXPERIMENTAL SETUP

We evaluate FairLoRA on three diverse datasets: CelebA, MultiNLI, and HateXplain, representing different modalities and bias types. For CelebA, we predict the "Male" attribute while accounting for "Blond Hair" as a sensitive attribute, revealing imbalances across male and female images with blond hair. In MultiNLI, we predict entailment relations with a focus on negation as a sensitive attribute, uncovering linguistic biases. HateXplain helps assess overlapping biases related to gender and race, focusing on hate speech prediction.

We compare FairLoRA with several widely-adopted baseline methods, including ERM, GroupDRO (Sagawa et al., 2020a), DFR (Kirichenko et al., 2023), and Lu et al. Lu et al. (2024), to demonstrate its effectiveness. FairLoRA is evaluated in two configurations:

- **FairLoRA Min.**: FairLoRA fine-tuning on the minority group to enhance fairness towards underrepresented groups, while ensuring no degradation in the overall model performance.

- **FairLoRA Maj.**: FairLoRA fine-tuning on the majority group to improve overall performance, while ensuring that the fairness for minority groups is not compromised.

FairLoRA's performance is evaluated using multiple metrics, including Accuracy (ACC), Balanced Accuracy (BA), Worst-Group Accuracy (WGA), Equalized Odds Difference (EOD), Demographic Parity (DP), Equal Opportunity (EOp), and Pearson Correlation Coefficient (PCC). All models are implemented using PyTorch, and we maintain consistent hyperparameter settings across experiments. Detailed implementation choices and hyperparameters can be found in the Appendix B.

## 5.2 EXPERIMENT 1: ELIMINATING BIAS OF A SINGLE TYPE

In our first experiment, we assess the effectiveness of FairLoRA in mitigating a single type of bias present in the CelebA and MultiNLI datasets. We used a 8-layer Vision Transformer (VIT) (Dosovitskiy, 2020) for the CelebA dataset and BERT-base (Devlin et al., 2019) for the MultiNLI dataset, aligning with prior benchmarks. Table 1 presents the performance comparison across different methods on both datasets. The results demonstrate several key findings:

Table 1: Performance comparison across different datasets and methods.

| Method | CelebA | | | MultiNLI | | |
|---|---|---|---|---|---|---|
| | ACC↑(%) | WGA↑(%) | EOD↓(%) | ACC↑(%) | WGA↑(%) | EOD↓(%) |
| **ERM** | 95.8 ± 0.1 | 77.9 ± 2.6 | 10.0 ± 1.7 | 82.6 ± 0.3 | 67.3 ± 2.6 | 12.5 ± 1.5 |
| + FL Min. | 95.8 ± 0.2 | **82.0 ± 2.2** | **8.5 ± 1.4** | 82.7 ± 0.4 | **71.0 ± 2.5** | **10.8 ± 1.4** |
| + FL Maj. | **95.9 ± 0.1** | 77.2 ± 2.8 | 10.0 ± 1.6 | **82.8 ± 0.2** | 66.8 ± 2.7 | 12.7 ± 1.5 |
| **GroupDRO** | 94.4 ± 0.5 | 87.4 ± 1.4 | 4.8 ± 0.6 | 80.8 ± 0.6 | 77.2 ± 1.2 | 5.9 ± 0.9 |
| + FL Min. | 94.4 ± 0.5 | **88.8 ± 1.5** | **4.7 ± 0.5** | 80.7 ± 0.8 | **78.3 ± 1.4** | **5.5 ± 0.8** |
| + FL Maj. | **94.7 ± 0.4** | 84.4 ± 1.1 | 5.9 ± 0.4 | **81.2 ± 0.5** | 75.0 ± 2.9 | 6.0 ± 1.2 |
| **DFR** | 94.3 ± 1.4 | 86.0 ± 2.0 | 7.7 ± 0.8 | 81.9 ± 0.4 | 74.1 ± 1.0 | 6.7 ± 0.8 |
| + FL Min. | 94.5 ± 1.2 | **87.8 ± 1.9** | **6.9 ± 0.8** | 81.9 ± 0.3 | **76.0 ± 1.0** | **6.3 ± 0.7** |
| + FL Maj. | **95.6 ± 0.1** | 83.3 ± 2.1 | 8.1 ± 1.3 | **82.1 ± 0.7** | 73.0 ± 2.1 | 6.8 ± 0.9 |
| **Lu et al.** | 95.4 ± 0.4 | 81.4 ± 4.8 | 8.3 ± 2.0 | 82.0 ± 0.2 | 72.8 ± 0.7 | 8.3 ± 0.6 |
| + FL Min. | 95.5 ± 0.4 | **86.8 ± 2.2** | **6.2 ± 0.7** | 82.0 ± 0.2 | **75.0 ± 0.6** | **7.5 ± 0.6** |
| + FL Maj. | **95.9 ± 0.3** | 80.4 ± 4.3 | 8.6 ± 1.7 | **82.5 ± 0.4** | 71.8 ± 1.5 | 8.4 ± 1.0 |

[*] Bold values indicate the best performance in each category. "FL" refers to FairLoRA.

**FairLoRA Minority Improves Fairness**: Across all baseline methods and both datasets, applying FairLoRA Minority leads to significant improvements in fairness metrics, specifically in WGA and EOD. For instance, in the ERM framework on CelebA, WGA increases from 77.9% to 82.0%, and EOD decreases from 10.0% to 8.5%. Similarly, on MultiNLI, WGA improves from 67.3% to 71.0%, and EOD decreases from 12.5% to 10.8%. These improvements indicate that by fine-tuning on minority group data, FairLoRA allows the model to better capture the characteristics of underrepresented groups, leading to more equitable performance.

**FairLoRA Majority Enhances Overall Accuracy**: Applying FairLoRA Majority results in slight improvements in overall accuracy across baseline methods. For example, in the ERM framework, ACC increases from 95.8% to 95.9% on CelebA and from 82.6% to 82.8% on MultiNLI. While the improvements in WGA and reductions in EOD are less pronounced compared to FairLoRA Minority, these results suggest that focusing on the majority group primarily enhances overall performance without significantly affecting fairness metrics.

**Synergy with Existing Debiasing Methods**: The combination of FairLoRA with other debiasing methods like GroupDRO and DFR further improves fairness metrics. For instance, GroupDRO + FairLoRA Minority on CelebA improves WGA from 87.4% to 88.8% and reduces EOD from 4.8% to 4.7%. This synergy illustrates that, even for models that have undergone prior bias mitigation, the incorporation of FairLoRA fine-tuning can further enhance fairness while preserving, or potentially slightly improving, the model's original performance.

### 5.3 EXPERIMENT 2: ELIMINATING BIASES OF MULTIPLE TYPES

In our second experiment, we evaluate the capability of FairLoRA to sequentially mitigate multiple biases. We utilize two pre-trained language models: DistilBERT-base (Sanh et al., 2019) and BERT-base. The procedure involves initial training with ERM, followed by sequential application of FairLoRA to mitigate racial bias (**FairLoRA African American**) and then gender bias (**FairLoRA Female**). Table 2 presents the results of this process, revealing several key findings:

Table 2: Performance and fairness comparison during progressive debiasing of sensitive attributes for DistilBERT-base and BERT-base.

| Metric | DistilBERT-base | | | BERT-base | | |
|---|---|---|---|---|---|---|
| | ERM | FLoRa Afr. | FLoRa Fe. | ERM | FLoRa Afr. | FLoRa Fe. |
| DP (R)↓ | 38.2 ± 1.4 | 33.7 ± 1.4 | 32.8 ± 1.1 | 27.1 ± 0.9 | 14.0 ± 1.0 | 12.4 ± 0.7 |
| EOp (R)↓ | 14.9 ± 1.1 | 14.2 ± 1.0 | 13.1 ± 1.0 | 13.0 ± 0.8 | 8.4 ± 1.1 | 7.2 ± 1.0 |
| **EOD(R)↓** | 26.5 ± 0.7 | 24.4 ± 0.6 | **23.0 ± 0.6** | 20.1 ± 0.4 | 11.2 ± 0.6 | **9.8 ± 0.5** |
| DP (G)↓ | 7.4 ± 1.3 | 7.6 ± 1.1 | 12.9 ± 2.2 | 7.6 ± 1.5 | 8.5 ± 1.4 | 7.6 ± 1.0 |
| EOp (G)↓ | 13.0 ± 0.5 | 13.0 ± 0.5 | 2.0 ± 2.1 | 18.2 ± 0.8 | 16.7 ± 0.4 | 8.8 ± 1.4 |
| **EOD(G)↓** | 11.3 ± 1.1 | 11.2 ± 0.7 | **7.4 ± 0.6** | 12.9 ± 1.1 | 12.6 ± 0.9 | **8.2 ± 0.4** |
| **ACC↑** | 79.5 ± 0.2 | 79.6 ± 0.2 | **79.7 ± 0.3** | **79.8 ± 0.3** | 79.6 ± 0.5 | 79.7 ± 0.4 |

[*] Bold values indicate the best performance in each category, while underlined values represent the second-best results. "R" refers to Race, and "G" refers to Gender.

**Effective Sequential Mitigation of Biases**: After applying FairLoRA Race, we observe a notable reduction in EOD (Race) for both models, with DistilBERT-base decreasing from 26.5% to 24.4%, and BERT-base from 20.1% to 11.2%. Notably, EOD (Gender) remains relatively stable in this phase, showing only slight changes (11.3% to 11.2% for DistilBERT-base and 12.9% to 12.6% for BERT-base). In the second stage, applying FairLoRA Female further reduces EOD (Gender), dropping from 11.2% to 7.4% for DistilBERT-base and from 12.6% to 8.2% for BERT-base. Importantly, these reductions in EOD (Gender) are achieved while retaining or improving EOD (Race), with DistilBERT-base decreasing from 24.4% to 23.0% and BERT-base from 11.2% to 9.8%.

**No Negative Interference**: The sequential application of FairLoRA demonstrates that mitigating a new bias does not negate the improvements achieved in earlier stages. This observation is crucial, as it suggests that FairLoRA effectively prevents catastrophic forgetting, a common issue when fine-tuning models on new tasks. We quantify this non-interference by calculating the correlation between performance changes across stages. Specifically, after mitigating gender bias, we compare the changes in metrics unrelated to gender before and after gender debiasing, relative to the original ERM model. For the DistilBERT and BERT models, the correlation coefficients are 0.97 and 0.99, respectively, indicating that addressing the new bias does not disrupt the gains made in previous bias mitigation stages. The corresponding calculation processes are provided in Appendix C.2.

### 5.4 EXPERIMENT 3: IMPACT OF DATASET PROPORTIONS ON FAIRLORA

This experiment evaluates the effect of varying training data sizes on the discriminator performance of FairLoRA, using the CelebA dataset and a 8-layer ViT, as summarized in Table 3. The discriminator was trained on data proportions ranging from 0.1% to 100%, with the findings as follows:

**Increased Dataset Size Enhances Discriminator Performance**: As the training dataset size increases, the TPR/FPR ratio shows significant improvement. Notably, the TPR/FPR ratio rises from

Table 3: Impact of Different Training Data Sizes on FairLoRA's Discriminator Performance

| Size | Num | TPR (%) | FPR (%) | TPR/FPR | ACC (%) | WGA (%) | EOD (%) |
|------|-----|---------|---------|---------|---------|---------|---------|
| ERM | - | - | - | - | 95.8 ± 0.1 | 77.9 ± 2.6 | 10.0 ± 1.7 |
| 0.1% | 163 | 77.0 ± 1.5 | 7.03 ± 0.5 | 10.9 ± 0.8 | 95.7 ± 0.4 | 80.1 ± 1.2 | 9.11 ± 0.9 |
| 0.5% | 813 | 85.0 ± 1.2 | 7.63 ± 0.6 | 11.1 ± 0.7 | 95.7 ± 0.4 | 80.6 ± 1.8 | 8.90 ± 2.0 |
| 1% | 1,627 | 88.3 ± 1.3 | 8.34 ± 0.7 | 10.6 ± 0.9 | 95.7 ± 0.3 | 80.6 ± 1.6 | 8.86 ± 1.8 |
| 5% | 8,134 | 93.2 ± 1.0 | 8.09 ± 0.6 | 11.5 ± 0.6 | 95.8 ± 0.4 | **82.2 ± 2.5** | **8.27 ± 2.3** |
| 10% | 16,269 | 94.1 ± 0.8 | 7.11 ± 0.5 | 13.2 ± 0.7 | 95.8 ± 0.2 | 80.6 ± 1.8 | 8.88 ± 0.9 |
| 50% | 81,344 | **94.9 ± 0.7** | 4.52 ± 0.4 | 21.0 ± 1.1 | 95.8 ± 0.3 | 81.1 ± 2.0 | 8.75 ± 1.2 |
| 100% | 162,688 | 94.1 ± 0.6 | **3.45 ± 0.3** | **27.2 ± 1.2** | **95.9 ± 0.1** | 82.0 ± 2.2 | 8.50 ± 1.4 |

10.9 for a 0.1% sample to 27.2 for a 100% sample, suggesting enhanced discriminatory power with increased data. The 100% training size yields the highest TPR/FPR ratio of 27.2, highlighting the discriminator's ability to differentiate biased and non-biased instances effectively.

**Condition for Maintaining Performance**: According to Theorem 1, the condition for maintaining performance without degradation is given by $\frac{(1-p)}{p}$, which represents the ratio of non-biased to biased classes. In this case, the ratio is $\frac{138,503}{24,267} = \mathbf{5.71}$. A higher TPR/FPR ratio indicates stronger discriminatory capability, which helps in achieving fairness improvements without negatively impacting model accuracy, as evidenced by the trend of improved metrics with increased dataset size.

**Effective Use of Limited Sensitive Attribute Labels**: FairLoRA performs well even with limited sensitive attribute labels. With just 0.1% of the labeled data, FairLoRA outperforms the baseline (ERM) in in terms of WRA and EOD metrics, showing its efficiency in enhancing fairness while requiring minimal data. As the training size increases, the model's accuracy remains stable, while the fairness metrics continue to improve. This observation underscores FairLoRA's ability to effectively mitigate biases without compromising overall performance, even in scenarios with limited access to sensitive attribute labels.

## 5.5 ABLATION STUDY

We conduct an ablation study to evaluate the impact of the group discriminator in FairLoRA using the HateXplain dataset. The study compares FairLoRA with LoRA (without the discriminator), focusing on changes in accuracy and fairness across training batches and thresholds.

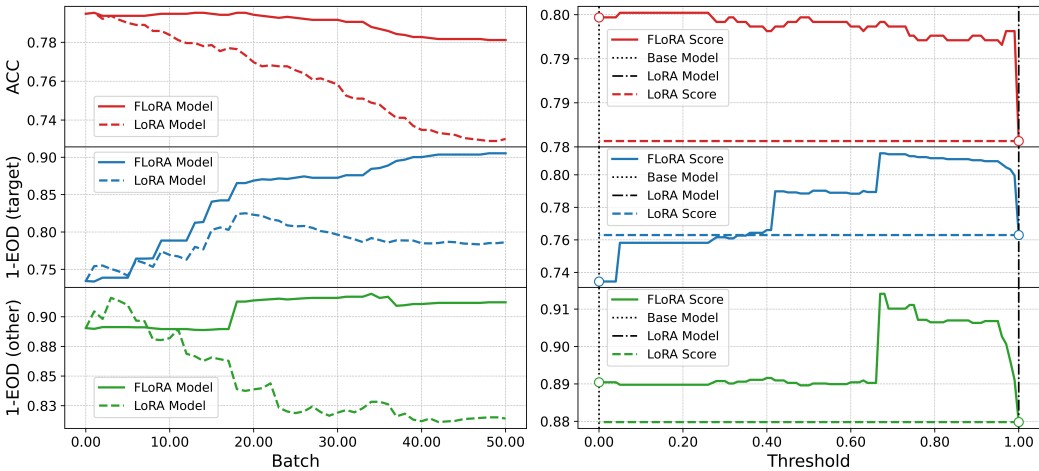

Figure 2: Comparison of accuracy (ACC) and fairness (1-EOD) between FairLoRA and LoRA. **Left**: Trends of ACC and fairness over training batches. **Right**: Impact of varying discriminator thresholds on ACC and fairness.

**Impact on Accuracy and Fairness Metrics:** Figure 2 (left) shows the accuracy (ACC) and fairness (1-EOD) trends across training batches. FairLoRA significantly improves debiasing for specific categories, such as African American-related comments, while maintaining model performance. Unlike LoRA, which shows a substantial decrease in accuracy from the early stages of training, FairLoRA exhibits minimal performance loss, maintaining stable accuracy even in later training stages. This demonstrates that FairLoRA can mitigate biases while preserving overall accuracy.

Regarding fairness towards non-target attributes (e.g., gender), FairLoRA maintains stable fairness throughout the training process, avoiding negative impacts on these attributes. In contrast, LoRA exhibits a significant decline in fairness for non-target attributes, suggesting that it struggles to ensure fairness across multiple sensitive categories when sequentially mitigating multiple types of biases. FairLoRA's ability to maintain relatively high $1 - EOD$ (other) scores indicates its robustness in handling multiple biases without catastrophic forgetting.

In target attribute fairness, FairLoRA consistently outperforms LoRA, with $1-EOD$ (target) improving gradually and staying at a high level throughout the training, while LoRA remains relatively low. This result demonstrates FairLoRA's superior capacity for enhancing fairness in bias mitigation.

**Effect of Discriminator Threshold:** Figure 2 (right) analyzes the effect of varying the discriminator threshold. A threshold of 0 corresponds to the base model, while a threshold of 1 represents the fully fine-tuned LoRA model. Across all thresholds, FairLoRA consistently outperforms LoRA in terms of accuracy. And with the increase of the threshold, FairLoRA's fairness in terms of EOD (target) continuously improves. Notably, when the threshold exceeds 0.4, FairLoRA achieves significantly better fairness for the target attribute ($1 - EOD$ (target)) than LoRA. Moreover, FairLoRA's fairness in terms of $1 - EOD$ (other) for non-target attributes also remains higher than that of LoRA, confirming that improving fairness in one category does not negatively impact other categories. Details on the TPR-to-FPR ratio variation are provided in Appendix D.

The ablation results confirm the crucial role of the group discriminator in FairLoRA, enabling superior fairness improvements while maintaining model performance. FairLoRA shows robustness across training iterations and threshold variations, significantly outperforming LoRA in both accuracy and fairness, particularly when addressing multiple types of biases sequentially.

## 6 CONCLUSIONS

In this article, we introduced FairLoRA, a bias mitigation method that employs discriminators with LoRA modules to enhance fairness while preserving model performance. Our experiments across various computer vision and natural language processing tasks demonstrate that FairLoRA can improve fairness metrics without compromising, and in some cases even enhancing, overall accuracy. FairLoRA showed consistent improvements in fairness across both single and multiple bias scenarios. It increased worst-group accuracy and reduced equalized odds difference in single bias settings, and effectively handled sequential debiasing of multiple biases (e.g., race and gender) without negatively impacting previous bias mitigation efforts. This highlights FairLoRA's robustness in handling multiple biases iteratively.

A key advantage of FairLoRA is its modular design, which enables targeted fine-tuning without the need for full-model training. This not only reduces computational costs but also minimizes the reliance on extensive sensitive attribute annotations, making FairLoRA highly adaptable to settings with partial information. Additionally, FairLoRA's selective activation of LoRA modules ensures that bias correction does not degrade overall model performance.

Beyond its applications in fairness, FairLoRA's modular approach has potential in multilingual and multi-task model optimization. For instance, in multilingual tasks, FairLoRA can be applied to fine-tune specific language components (e.g., improving Chinese language understanding) without impacting performance on other languages (e.g., English). Similarly, in multi-task settings, LoRA modules can be independently fine-tuned for specialized tasks, such as code generation or mathematical reasoning, without disrupting the model's core capabilities across other tasks. This flexibility enables FairLoRA to support the growing demands for adaptable, task-specific model training in diverse and multilingual environments, providing a pathway for improving both fairness and task performance.

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

## A   THEOREM PROOF

**Lemma 1.** For the majority group (G1), the performance of the model after FairLoRA fine-tuning is:

$$P(M_{\text{FairLoRA}}, \text{G1}) = (1 - \text{FPR}) \cdot P(M, \text{G1}) + \text{FPR} \cdot P(M_{\text{LoRA}}, \text{G1})$$

*Proof.*

*Definitions and Notations:*

- $M$: the original model.
- $M_{\text{LoRA}}$: the model fine-tuned using LoRA.
- $M_{\text{FairLoRA}}$: the final model after applying FairLoRA fine-tuning.
- G1: the majority group.
- $P(M, \text{G1})$: performance of model $M$ on group G1.
- FPR: False Positive Rate when predicting G2 for samples from G1.

In the context of FairLoRA fine-tuning, the performance of the model on G1 depends on how samples from G1 are classified:

- *True Negatives (TN)*: samples from G1 correctly classified as G1.
- *False Positives (FP)*: samples from G1 incorrectly classified as G2.

*Calculating the Performance:*

Let $N_1$ be the total number of samples in G1.

- Number of True Negatives: $\text{TN} = (1 - \text{FPR}) \cdot N_1$.
- Number of False Positives: $\text{FP} = \text{FPR} \cdot N_1$.

For G1, the FairLoRA model uses:

- The original model $M$ for True Negatives.
- The LoRA fine-tuned model $M_{\text{LoRA}}$ for False Positives.

Thus, the total performance on G1 is the weighted average:

$$\begin{aligned}
P(M_{\text{FairLoRA}}, \text{G1}) &= \frac{\text{Performance on TN} + \text{Performance on FP}}{N_1} \\
&= \frac{\text{TN} \cdot P(M, \text{G1}) + \text{FP} \cdot P(M_{\text{LoRA}}, \text{G1})}{N_1} \\
&= \frac{[(1 - \text{FPR})N_1 P(M, \text{G1}) + \text{FPR} N_1 P(M_{\text{LoRA}}, \text{G1})]}{N_1} \\
&= (1 - \text{FPR}) \cdot P(M, \text{G1}) + \text{FPR} \cdot P(M_{\text{LoRA}}, \text{G1}).
\end{aligned}$$

Therefore, we have:

$$P(M_{\text{FairLoRA}}, \text{G1}) = (1 - \text{FPR}) \cdot P(M, \text{G1}) + \text{FPR} \cdot P(M_{\text{LoRA}}, \text{G1}).$$

This completes the proof. □

**Lemma 2.** For the minority group (G2), the performance of the model after FairLoRA fine-tuning is:

$$P(M_{\text{FairLoRA}}, \text{G2}) = \text{TPR} \cdot P(M_{\text{LoRA}}, \text{G2}) + (1 - \text{TPR}) \cdot P(M, \text{G2})$$

*Proof.*

*Definitions and Notations:*

- G2: the minority group.
- $P(M, \text{G2})$: performance of model $M$ on group G2.
- TPR: True Positive Rate when correctly predicting G2 for samples from G2.

For samples from G2, their classification can be:

- *True Positives (TP)*: samples from G2 correctly classified as G2.
- *False Negatives (FN)*: samples from G2 incorrectly classified as G1.

*Calculating the Performance:*

Let $N_2$ be the total number of samples in G2.

- Number of True Positives: $\text{TP} = \text{TPR} \cdot N_2$.
- Number of False Negatives: $\text{FN} = (1 - \text{TPR}) \cdot N_2$.

For G2, the FairLoRA model uses:

- The LoRA fine-tuned model $M_{\text{LoRA}}$ for True Positives.
- The original model $M$ for False Negatives.

Thus, the total performance on G2 is:

$$
\begin{aligned}
P(M_{\text{FairLoRA}}, \text{G2}) &= \frac{\text{Performance on TP} + \text{Performance on FN}}{N_2} \\
&= \frac{\text{TP} \cdot P(M_{\text{LoRA}}, \text{G2}) + \text{FN} \cdot P(M, \text{G2})}{N_2} \\
&= \frac{[\text{TPR} N_2 P(M_{\text{LoRA}}, \text{G2}) + (1 - \text{TPR}) N_2 P(M, \text{G2})]}{N_2} \\
&= \text{TPR} \cdot P(M_{\text{LoRA}}, \text{G2}) + (1 - \text{TPR}) \cdot P(M, \text{G2}).
\end{aligned}
$$

Therefore, we have:

$$
P(M_{\text{FairLoRA}}, \text{G2}) = \text{TPR} \cdot P(M_{\text{LoRA}}, \text{G2}) + (1 - \text{TPR}) \cdot P(M, \text{G2}).
$$

This completes the proof. $\qquad\square$

**Theorem 1.** To ensure that FairLoRA does not degrade the overall performance of the model, the ratio of the true positive rate (TPR) to the false positive rate (FPR) must satisfy:

$$
\frac{\text{TPR}}{\text{FPR}} \geq \frac{(1 - p)}{p} \cdot \frac{P(M, \text{G1}) - P(M_{\text{LoRA}}, \text{G1})}{P(M_{\text{LoRA}}, \text{G2}) - P(M, \text{G2})}
$$

*Proof.*

*Definitions and Notations:*

- $p = \frac{N_2}{N_1 + N_2}$: proportion of samples from G2.
- $(1 - p)$: proportion of samples from G1.
- $\Delta P(\text{G1})$: change in performance on G1.
- $\Delta P(\text{G2})$: change in performance on G2.
- $\Delta P$: overall change in performance.

*Calculating the Change in Performance for G1:*

From Theorem 1, the performance change on G1 is:

$$
\begin{aligned}
\Delta P(\text{G1}) &= P(M_{\text{FairLoRA}}, \text{G1}) - P(M, \text{G1}) \\
&= [(1 - \text{FPR}) P(M, \text{G1}) + \text{FPR} P(M_{\text{LoRA}}, \text{G1})] - P(M, \text{G1}) \\
&= -\text{FPR} \cdot P(M, \text{G1}) + \text{FPR} \cdot P(M_{\text{LoRA}}, \text{G1}) \\
&= \text{FPR} \cdot [P(M_{\text{LoRA}}, \text{G1}) - P(M, \text{G1})].
\end{aligned}
$$

*Calculating the Change in Performance for G2:*

From Theorem 2, the performance change on G2 is:

$$
\begin{aligned}
\Delta P(\text{G2}) &= P(M_{\text{FairLoRA}}, \text{G2}) - P(M, \text{G2}) \\
&= [\text{TPR}P(M_{\text{LoRA}}, \text{G2}) + (1 - \text{TPR})P(M, \text{G2})] - P(M, \text{G2}) \\
&= -\text{TPR} \cdot P(M, \text{G2}) + \text{TPR} \cdot P(M_{\text{LoRA}}, \text{G2}) \\
&= \text{TPR} \cdot [P(M_{\text{LoRA}}, \text{G2}) - P(M, \text{G2})].
\end{aligned}
$$

*Calculating the Overall Change in Performance:*

The overall change is the weighted sum:

$$
\Delta P = (1 - p) \cdot \Delta P(\text{G1}) + p \cdot \Delta P(\text{G2}).
$$

Substituting the expressions for $\Delta P(\text{G1})$ and $\Delta P(\text{G2})$:

$$
\Delta P = (1 - p) \cdot \text{FPR}[P(M_{\text{LoRA}}, \text{G1}) - P(M, \text{G1})] + p \cdot \text{TPR}[P(M_{\text{LoRA}}, \text{G2}) - P(M, \text{G2})].
$$

*Setting the Condition for No Performance Degradation:*

To ensure the overall performance does not degrade ($\Delta P \geq 0$), we require:

$$
(1 - p) \cdot \text{FPR}[P(M_{\text{LoRA}}, \text{G1}) - P(M, \text{G1})] + p \cdot \text{TPR}[P(M_{\text{LoRA}}, \text{G2}) - P(M, \text{G2})] \geq 0.
$$

*Assuming Performance Changes:*

- Let $\Delta P_{\text{G1}} = P(M_{\text{LoRA}}, \text{G1}) - P(M, \text{G1})$ (likely negative).
- Let $\Delta P_{\text{G2}} = P(M_{\text{LoRA}}, \text{G2}) - P(M, \text{G2})$ (positive).

Rewriting the inequality:

$$
(1 - p) \cdot \text{FPR} \cdot \Delta P_{\text{G1}} + p \cdot \text{TPR} \cdot \Delta P_{\text{G2}} \geq 0.
$$

*Solving for $\frac{TPR}{FPR}$:*

1. Isolate the positive term:
$$
p \cdot \text{TPR} \cdot \Delta P_{\text{G2}} \geq -(1 - p) \cdot \text{FPR} \cdot \Delta P_{\text{G1}}.
$$

2. Since $\Delta P_{\text{G1}} < 0$, $-\Delta P_{\text{G1}} > 0$:
$$
p \cdot \text{TPR} \cdot \Delta P_{\text{G2}} \geq (1 - p) \cdot \text{FPR} \cdot (-\Delta P_{\text{G1}}).
$$

3. Divide both sides by $p \cdot \Delta P_{\text{G2}}$ (which is positive):
$$
\text{TPR} \geq \frac{(1 - p)}{p} \cdot \frac{\text{FPR} \cdot (-\Delta P_{\text{G1}})}{\Delta P_{\text{G2}}}.
$$

4. Divide both sides by FPR (assuming FPR $> 0$):
$$
\frac{\text{TPR}}{\text{FPR}} \geq \frac{(1 - p)}{p} \cdot \frac{-\Delta P_{\text{G1}}}{\Delta P_{\text{G2}}}.
$$

5. Substitute back the definitions of $\Delta P_{\text{G1}}$ and $\Delta P_{\text{G2}}$:
$$
\frac{\text{TPR}}{\text{FPR}} \geq \frac{(1 - p)}{p} \cdot \frac{P(M, \text{G1}) - P(M_{\text{LoRA}}, \text{G1})}{P(M_{\text{LoRA}}, \text{G2}) - P(M, \text{G2})}.
$$

Therefore, the ratio of the True Positive Rate to the False Positive Rate must satisfy:

$$
\frac{\text{TPR}}{\text{FPR}} \geq \frac{(1 - p)}{p} \cdot \frac{P(M, \text{G1}) - P(M_{\text{LoRA}}, \text{G1})}{P(M_{\text{LoRA}}, \text{G2}) - P(M, \text{G2})}.
$$

This condition ensures that the positive impact on G2 outweighs the negative impact on G1, preventing overall performance degradation.

This completes the proof. $\qquad\square$

# B IMPLEMENTATION DETAILS OF FAIRLORA

This section provides the implementation details for FairLoRA, focusing on the group discriminator training, fine-tuning dataset construction, and FairLoRA training configuration. Key components of the implementation are presented in pseudocode to facilitate understanding and reproducibility.

**Group Discriminator Training**

To effectively identify sensitive attributes, we trained a group discriminator $D_\phi$ that takes hidden layer representations from a pre-trained model as input and outputs the corresponding sensitive attribute labels. Specifically, we used the penultimate hidden states $h_\theta(x) \in \mathbb{R}^{T \times d}$ as input, where $T$ represents the sequence length and $d$ is the dimensionality of the hidden states.

To aggregate the sequence representation into a global vector, we employed attention pooling, which assigns importance weights to different time steps. This allows the model to focus on the most relevant parts of the sequence when predicting sensitive attributes.

To mitigate bias in predicting sensitive attributes, we employed the worst-group cross-entropy loss:

$$\mathcal{L}_{\text{worst}} = \max_{g \in \mathcal{G}} \mathbb{E}_{(x,s) \sim P_g} \left[ \ell \left( D_\phi(h_{\text{pool}}(x)), s \right) \right],$$

where $\mathcal{G}$ represents the set of all groups, $P_g$ is the data distribution for group $g$, $s$ is the sensitive attribute label, and $\ell(\cdot)$ denotes the cross-entropy loss function.

The combined pseudocode for the attention pooling mechanism and the group discriminator network is presented below.

---

**Algorithm 1** Group Discriminator with Attention Pooling

---

**Require:** Hidden states $h \in \mathbb{R}^{T \times d}$
**Ensure:** Predicted sensitive attribute label $\hat{s}$
1: **Attention Pooling:**
2: Initialize learnable parameter vector $w \in \mathbb{R}^d$
3: **for** $t = 1$ to $T$ **do**
4:     Compute attention score: $a_t \leftarrow w^\top h_t$         ▷ Scalar value
5: **end for**
6: Compute attention weights: $\alpha \leftarrow \text{softmax}([a_1, a_2, \ldots, a_T])$
7: Compute pooled representation: $h_{\text{pool}} \leftarrow \sum_{t=1}^{T} \alpha_t h_t$

8: **Group Discriminator Network:**
9: Compute hidden layer activation: $z \leftarrow \text{ReLU}(W_1 h_{\text{pool}} + b_1)$     ▷ $W_1 \in \mathbb{R}^{d_1 \times d}$
10: Compute output logits: $o \leftarrow W_2 z + b_2$     ▷ $W_2 \in \mathbb{R}^{2 \times d_1}$
11: Compute predicted probabilities: $\hat{p} \leftarrow \text{sigmoid}(o)$
12: Predict sensitive attribute: $\hat{s} \leftarrow \arg \max \hat{p}$

---

In this algorithm:

Attention Pooling (Lines 2–7): We compute attention scores for each time step using the learnable parameter vector $w$. The attention weights $\alpha$ are obtained by applying the softmax function to the attention scores. The pooled representation $h_{\text{pool}}$ is then calculated as the weighted sum of the hidden states.

Group Discriminator Network (Lines 8–12): The pooled representation $h_{\text{pool}}$ is fed into a fully connected layer with ReLU activation to obtain the hidden activation $z$. A second linear layer computes the logits $o$, which are transformed into probabilities $\hat{p}$ using the sigmoid function. The predicted sensitive attribute label $\hat{s}$ is determined by taking the class with the highest probability.

By combining the attention pooling mechanism with the group discriminator network in a single algorithm, we provide a clear and concise representation of how the discriminator processes the input hidden states to predict sensitive attributes.

Using all available data in these experiments ensures that the discriminators achieve high accuracy, thereby improving the model's capacity to debias effectively without compromising performance. For scenarios with limited sensitive attribute labels, results are presented separately in Table 3.

**Partition of dataset**

For CelebA and MultiNLI, we used the official splits provided in the respective documentation, following the standard training and test set divisions. For HateXplain, since the official split is not provided, we followed the approach of Lu et al.Lu et al. (2024), where 50% of the samples were used as the test set.

**FairLoRA Fine-tuning Dataset Construction**

The fine-tuning dataset was constructed to ensure class balance through the following steps:

- **Data with Sensitive Attribute Labels:** We selected samples with a sensitive attribute label of $s = 1$ and performed undersampling to balance the classes.

- **Data without Sensitive Attribute Labels:** A trained discriminator $D_\phi$ was used to assign pseudo-labels for sensitive attributes. Samples predicted as $s = 1$ were selected, and undersampling was applied to balance the class distribution.

**FairLoRA Training Configuration**

We employed the AdamW optimizer for training, which effectively handles weight decay and improves generalization. The learning rate was set to $1 \times 10^{-5}$ to ensure stable convergence during fine-tuning. Training was conducted for 2 epochs, as this was sufficient for the model to converge without overfitting. To maintain consistency, $\tau$ was fixed at 0.5 across all experiments. Additionally, we used five different random seeds (5, 15, 25, 35, 45) for each set of experiments to ensure robustness. A validation set can also be utilized to guide hyperparameter tuning if needed.

**Pseudocode Implementation**

During training, all LoRA adjustments are retained to allow the model to fully learn from the FairLoRA fine-tuning dataset. During inference, the discriminator's output selectively activates the LoRA adjustments for samples predicted as belonging to sensitive groups. This design ensures that model adjustments are targeted to reduce bias where needed, while maintaining both efficiency and overall performance.

FairLoRA can be extended to accommodate multiple sensitive attributes by introducing additional discriminators and LoRA modules.

---

**Algorithm 2** FairLoRA Forward Pass with Multiple Sensitive Attributes

---

**Require:** Input features $x$, discriminator outputs $\texttt{dis}_1, \texttt{dis}_2, \ldots, \texttt{dis}_k$, training mode flag $\texttt{training}$
1: Compute base output: $y_{\text{base}} \leftarrow \texttt{LinearLayer}(x)$
2: **for** $i = 1$ to $k$ **do**
3:      Compute LoRA adjustment: $y_{\text{lora}_i} \leftarrow \texttt{LoRALayer}_i(x)$
4:      Determine if $\text{LoRA}_i$ should be applied: $\text{apply\_lora}_i \leftarrow \texttt{dis}_i > \tau$
5:      **if not** $\texttt{training}$ **then**
6:          $y_{\text{lora}_i}[\neg\text{apply\_lora}_i] \leftarrow 0$
7:      **end if**
8: **end for**
9: **return** $y \leftarrow y_{\text{base}} + \sum_{i=1}^{k} y_{\text{lora}_i}$

---

This approach enhances the fairness of the model without requiring full access to all sensitive attribute labels, ensuring fairer treatment of underrepresented groups while preserving overall performance.

# C COMPREHENSIVE COMPARISON OF EXPERIMENTAL DATA

The evaluation metrics employed in the presented tables are critical for assessing both the performance and fairness of the models:

- **Accuracy (ACC)**: The overall proportion of correctly predicted instances among all samples.

- **Balanced Accuracy (BA)**: Accounts for class imbalance by computing the average recall obtained on each class. It is calculated as:

$$BA = \frac{1}{2} \left( \frac{TP}{TP + FN} + \frac{TN}{TN + FP} \right)$$

  where TP, TN, FP, and FN denote true positives, true negatives, false positives, and false negatives, respectively.

- **Worst Group Accuracy (WGA)**: The lowest accuracy observed among all evaluated groups (e.g., different genders, races), highlighting the model's performance on the most disadvantaged group.

- **Demographic Parity (DP)**: Measures the difference in positive prediction rates across different groups. A lower DP indicates more equitable positive prediction distributions among groups.

- **Equal Opportunity (EOp)**: Assesses the disparity in true positive rates (TPR) between groups. A smaller EOp suggests that the model provides similar chances of correct positive predictions across groups.

- **Equalized Odds Difference (EOD)**: Considers both TPR and false positive rate (FPR) differences between groups. Lower EOD values indicate more balanced predictive performance across groups in terms of both positive and negative classes.

- **Average Error Rate (AER)**: The mean error rate across different groups. A lower AER signifies an overall reduction in model errors.

## C.1 COMPARATIVE ANALYSIS OF DEBIASING FOR SINGLE SENSITIVE ATTRIBUTE

The analysis of Table 4 involves evaluating the performance and fairness metrics of different models on the CelebA dataset.

Table 4: Performance and Fairness Metrics of Models on the CelebA Dataset

| Model | ACC↑(%) | BA↑(%) | WGA↑(%) | DP↓(%) | EOp↓(%) | EOD↓(%) | AER↑(%) |
|---|---|---|---|---|---|---|---|
| **ERM** | 95.8 ± 0.1 | 95.7 ± 0.0 | 77.9 ± 2.6 | 37.1 ± 0.6 | 17.5 ± 2.9 | 10.0 ± 1.7 | 69.7 ± 3.9 |
| + FL Min. | 95.8 ± 0.2 | **95.8 ± 0.1** | **82.0 ± 2.2** | 37.3 ± 0.5 | **14.2 ± 2.4** | **8.5 ± 1.4** | 68.7 ± 2.9 |
| + FL Maj. | **95.9 ± 0.1** | 95.6 ± 0.1 | 77.2 ± 2.8 | **36.9 ± 0.6** | 17.8 ± 3.0 | 10.0 ± 1.6 | 67.8 ± 3.0 |
| + FL All | **95.9 ± 0.1** | **95.8 ± 0.1** | 81.3 ± 1.5 | 37.1 ± 0.3 | 14.6 ± 1.7 | 8.6 ± 1.0 | **70.3 ± 4.1** |
| **GroupDRO** | 94.4 ± 0.5 | 94.4 ± 0.4 | 87.4 ± 1.4 | **35.1 ± 0.5** | 7.5 ± 1.2 | 4.8 ± 0.6 | 81.8 ± 6.7 |
| + FL Min. | 94.4 ± 0.5 | 94.6 ± 0.4 | **88.8 ± 1.5** | 35.4 ± 0.4 | **6.8 ± 1.3** | **4.7 ± 0.5** | **83.3 ± 6.7** |
| + FL Maj. | **94.7 ± 0.4** | 94.6 ± 0.4 | 84.4 ± 1.1 | 35.4 ± 0.3 | 9.7 ± 0.9 | 5.9 ± 0.4 | 72.1 ± 3.6 |
| + FL All | **94.7 ± 0.3** | **94.7 ± 0.3** | 85.9 ± 1.6 | 35.6 ± 0.2 | 9.0 ± 1.6 | 5.7 ± 0.8 | 75.1 ± 6.1 |
| **DFR** | 94.3 ± 1.4 | 94.8 ± 1.0 | 86.0 ± 2.0 | 37.5 ± 0.6 | 11.1 ± 1.6 | 7.7 ± 0.8 | 75.1 ± 4.4 |
| + FL Min. | 94.5 ± 1.2 | 95.0 ± 0.9 | **87.8 ± 1.9** | 37.4 ± 0.8 | **9.6 ± 1.3** | 6.9 ± 0.8 | **78.7 ± 8.4** |
| + FL Maj. | **95.6 ± 0.1** | **95.7 ± 0.0** | 83.3 ± 2.1 | **37.2 ± 0.5** | 13.1 ± 2.3 | 8.1 ± 1.3 | 72.3 ± 5.5 |
| + FL All | 95.4 ± 0.1 | **95.7 ± 0.1** | 86.0 ± 1.1 | 37.3 ± 0.3 | 11.0 ± 1.2 | 7.1 ± 0.6 | 74.5 ± 6.1 |
| **Lu et al. (2024)** | 95.4 ± 0.4 | 95.6 ± 0.4 | 81.4 ± 4.8 | 36.8 ± 0.5 | 14.1 ± 4.1 | 8.3 ± 2.0 | 68.7 ± 5.3 |
| + FL Min. | 95.5 ± 0.4 | 95.7 ± 0.3 | **86.8 ± 2.2** | **36.7 ± 0.5** | **9.8 ± 1.6** | **6.2 ± 0.7** | **75.9 ± 8.2** |
| + FL Maj. | **95.9 ± 0.3** | 95.7 ± 0.3 | 80.4 ± 4.3 | **36.7 ± 0.6** | 14.8 ± 3.5 | 8.6 ± 1.7 | 67.3 ± 4.5 |
| + FL All | 95.6 ± 0.3 | **95.8 ± 0.2** | 86.6 ± 2.1 | **36.7 ± 0.7** | 10.0 ± 1.4 | 6.3 ± 0.6 | 75.7 ± 9.0 |

[*] Bold values indicate the best performance in each category.

The **ERM model** achieves high overall accuracy (ACC) and balanced accuracy (BA), with scores of approximately 95.8% and 95.7%, respectively. However, the model presents fairness concerns as indicated by the worst-group accuracy (WGA), which is relatively low at 77.9%. This suggests suboptimal performance for the least advantaged group. Incorporating the **FL Min.** strategy increases the WGA to 82.0%, demonstrating improved performance on the worst-performing group. Additionally, there is a reduction in the Equal Opportunity (EOp) metric from 17.5% to 14.2% and in Equalized Odds Difference (EOD) from 10.0% to 8.5%, indicating a significant decrease in group disparities and an overall enhancement in fairness.

The **GroupDRO model** initially performs well with a high WGA of 87.4%, reflecting strong baseline performance for the worst-performing group. When **FL Min.** is applied, the WGA further increases to 88.8%, enhancing the model's robustness across groups. Moreover, there are decreases in EOp from 7.5% to 6.8% and in EOD from 4.8% to 4.7%, implying a reduction in group disparities and improved fairness metrics.

The **DFR model** attains a WGA of 86.0%, suggesting favorable fairness performance at the baseline level. With the application of **FL Min.**, the WGA improves to 87.8%, indicating better performance on the worst-performing group. Concurrently, the EOp decreases from 11.1% to 9.6%, and the EOD reduces from 7.7% to 6.9%, which enhances fairness by mitigating disparities between different groups.

The **Lu et al. (2024) model** starts with a WGA of 81.4%, highlighting room for improvement in addressing the worst-performing group. Upon incorporating **FL Min.**, the WGA significantly increases to 86.8%, indicating substantial improvement for disadvantaged groups. Additionally, notable reductions are observed in EOp from 14.1% to 9.8%, and in EOD from 8.3% to 6.2%, demonstrating enhanced fairness by reducing inter-group disparities.

Table 5: Performance comparison across different attributes of CelebA dataset.

| Method | Heavy Makeup | | | Wearing Lipstick | | |
|---|---|---|---|---|---|---|
| | ACC↑(%) | WGA↑(%) | EOD↓(%) | ACC↑(%) | WGA↑(%) | EOD↓(%) |
| **ERM** | 95.8 ± 0.1 | 45.4 ± 3.2 | 27.9 ± 1.9 | 95.8 ± 0.1 | 57.4 ± 3.5 | 29.3 ± 2.4 |
| + FL Min. | 95.8 ± 0.1 | **54.5 ± 3.1** | **24.4 ± 1.7** | 95.8 ± 0.2 | **63.0 ± 2.7** | **25.1 ± 2.0** |
| **GroupDRO** | 94.4 ± 0.5 | 65.4 ± 2.7 | 25.8 ± 1.6 | 94.4 ± 0.5 | 70.2 ± 2.5 | 25.9 ± 1.9 |
| + FL Min. | 94.4 ± 0.4 | **70.1 ± 2.5** | **22.7 ± 1.5** | **94.5 ± 0.4** | **74.3 ± 2.4** | **22.5 ± 1.9** |
| **DFR** | 94.3 ± 1.4 | 58.0 ± 2.2 | 27.0 ± 1.8 | 94.3 ± 1.4 | 68.1 ± 1.9 | 26.7 ± 1.8 |
| + FL Min. | **94.5 ± 1.5** | **63.8 ± 1.9** | **24.1 ± 2.0** | **94.4 ± 1.4** | **73.2 ± 2.0** | **22.3 ± 1.7** |
| **Lu et al.** | 95.4 ± 0.4 | 61.4 ± 2.5 | 28.0 ± 2.2 | 95.4 ± 0.4 | 67.8 ± 2.1 | 27.5 ± 1.7 |
| + FL Min. | **95.6 ± 0.5** | **69.8 ± 2.9** | **23.2 ± 2.5** | 95.4 ± 0.4 | **74.1 ± 2.3** | **23.1 ± 1.5** |

We also conducted experiments using other sensitive attributes, such as "Heavy Makeup" and "Wearing Lipstick". The results, presented in Table 5, are consistent with those in Table 4, demonstrating the robustness of our proposed method.

The analysis of Table 6, which presents the performance and fairness metrics of models on the MultiNLI dataset, follows a similar structure to that of Table 1. The general observations about model performance and the impact of incorporating fairness learning strategies (such as FL Min., FL Maj., and FL All) are consistent with the results discussed for the CelebA dataset.

In summary, incorporating the **FL Min.** strategy across all models for the MultiNLI dataset leads to similar improvements as observed with the CelebA dataset. The WGA increases, and the fairness disparities (as indicated by DP, EOp, and EOD) are reduced. These results emphasize that focusing on disadvantaged groups during model training enhances both the performance for those groups and overall fairness.

Table 6: Performance and Fairness Metrics of Models on the MultiNLI Dataset

| Model | ACC↑(%) | BA↑(%) | WGA↑(%) | DP↓(%) | EOp↓(%) | EOD↓(%) | AER↑(%) |
|---|---|---|---|---|---|---|---|
| **ERM** | 82.6 ± 0.3 | 82.6 ± 0.3 | 67.3 ± 2.6 | 47.6 ± 1.2 | 14.6 ± 1.1 | 12.5 ± 1.5 | 57.1 ± 4.0 |
| + FL Min. | 82.7 ± 0.4 | 82.7 ± 0.4 | **71.0 ± 1.5** | **45.5 ± 0.7** | **12.2 ± 1.0** | **10.8 ± 1.4** | **60.2 ± 3.8** |
| + FL Maj. | **82.8 ± 0.2** | **82.8 ± 0.2** | 66.8 ± 2.7 | 47.7 ± 1.4 | 14.7 ± 1.2 | 12.7 ± 1.5 | 55.6 ± 4.1 |
| + FL All | **82.8 ± 0.2** | **82.8 ± 0.2** | 70.5 ± 2.2 | 45.8 ± 1.1 | 12.5 ± 1.1 | 11.0 ± 1.0 | 59.0 ± 4.0 |
| **GroupDRO** | 80.8 ± 0.6 | 80.8 ± 0.3 | 77.2 ± 1.2 | 40.7 ± 0.4 | 8.8 ± 0.7 | 5.9 ± 0.9 | 74.8 ± 6.5 |
| + FL Min. | 80.7 ± 0.8 | 80.7 ± 0.8 | **78.3 ± 1.4** | **39.6 ± 0.7** | **7.5 ± 0.6** | **5.5 ± 0.8** | **77.2 ± 7.1** |
| + FL Maj. | **81.2 ± 0.5** | **81.2 ± 0.5** | 75.0 ± 2.9 | 42.5 ± 0.6 | 9.1 ± 0.7 | 6.0 ± 1.2 | 72.5 ± 5.6 |
| + FL All | **81.2 ± 0.4** | **81.2 ± 0.4** | 76.8 ± 1.0 | 41.6 ± 0.9 | 8.3 ± 0.8 | 5.7 ± 0.9 | 74.9 ± 6.7 |
| **DFR** | 81.9 ± 0.4 | 81.9 ± 0.4 | 74.1 ± 1.0 | 43.1 ± 0.5 | 9.1 ± 0.7 | 6.7 ± 0.8 | 65.1 ± 5.2 |
| + FL Min. | 81.9 ± 0.3 | 81.9 ± 0.3 | **76.0 ± 1.0** | **42.0 ± 0.4** | **8.0 ± 0.6** | **6.3 ± 0.7** | 67.3 ± 5.4 |
| + FL Maj. | **82.1 ± 0.7** | **82.1 ± 0.7** | 73.0 ± 2.1 | 43.9 ± 0.8 | 9.0 ± 1.0 | 6.8 ± 0.9 | **63.4 ± 7.1** |
| + FL All | **82.1 ± 0.5** | **82.1 ± 0.5** | 74.7 ± 1.5 | 42.9 ± 0.5 | 8.5 ± 0.7 | 6.6 ± 0.7 | 66.0 ± 6.0 |
| **Lu et al. (2024)** | 82.0 ± 0.2 | 82.0 ± 0.2 | 72.8 ± 0.7 | 44.7 ± 0.9 | 10.1 ± 0.6 | 8.3 ± 0.6 | 64.7 ± 5.1 |
| + FL Min. | 82.0 ± 0.2 | 82.0 ± 0.2 | **75.0 ± 0.6** | **42.6 ± 0.8** | **9.0 ± 0.5** | **7.5 ± 0.6** | **66.9 ± 5.2** |
| + FL Maj. | 82.5 ± 0.4 | 82.5 ± 0.4 | 71.8 ± 1.5 | 44.8 ± 1.2 | 10.7 ± 1.2 | 8.4 ± 1.0 | 62.7 ± 6.7 |
| + FL All | **82.6 ± 0.1** | **82.6 ± 0.1** | 74.7 ± 0.6 | 43.1 ± 0.9 | 9.1 ± 0.6 | 7.7 ± 0.6 | 66.3 ± 5.0 |

## C.2 CALCULATION OF CORRELATION COEFFICIENTS

To verify that mitigating a new bias does not interfere with previously achieved fairness improvements, we calculated the Pearson correlation coefficients between performance changes across debiasing stages. Specifically, we examined the changes in metrics unrelated to gender bias after mitigating gender bias, relative to the original ERM model. The following metrics were used for each model: **DP (R)** (racial fairness), **EOp (R)**, **EOD (R)** and **ACC** (accuracy).

**1. Extract Metrics and Compute Changes**

The metrics were extracted from Table 7. For each metric $M$, we calculated the change $\Delta M$ at each debiasing stage relative to the ERM baseline.

For DistilBERT-base, the changes are:

- **Changes at FLoRa Afr. stage**: $\Delta DP (R)_{Afr.} = 33.7 - 38.2 = -4.5$, $\Delta EOp (R)_{Afr.} = 14.2 - 14.9 = -0.7$, $\Delta EOD (R)_{Afr.} = 24.4 - 26.5 = -2.1$, $\Delta ACC_{Afr.} = 79.6 - 79.5 = +0.1$.

- **Changes at FLoRa Fe. stage**: $\Delta DP (R)_{Fe.} = 32.8 - 38.2 = -5.4$, $\Delta EOp (R)_{Fe.} = 13.1 - 14.9 = -1.8$, $\Delta EOD (R)_{Fe.} = 23.0 - 26.5 = -3.5$, $\Delta ACC_{Fe.} = 79.7 - 79.5 = +0.2$.

**2. Form Vectors of Changes**

We form vectors of the changes for the two debiasing stages: $\mathbf{X} = [-4.5, -0.7, -2.1, +0.1]$ (FLoRa Afr.), $\mathbf{Y} = [-5.4, -1.8, -3.5, +0.2]$ (FLoRa Fe.).

**3. Compute Correlation Coefficient**

The Pearson correlation coefficient $r$ between the vectors $\mathbf{X}$ and $\mathbf{Y}$ was calculated. For DistilBERT-base, the resulting correlation coefficient is:

$$r = 0.97$$

**4. Results for BERT-base Model**

Similarly, for the BERT-base model, we calculated:

- **Changes at FLoRa Afr. and FLoRa Fe. stages**: $\mathbf{X} = [-13.1, -4.6, -8.9, -0.2]$ (FLoRa Afr.), $\mathbf{Y} = [-14.7, -5.8, -10.3, -0.1]$ (FLoRa Fe.).

- **Correlation Coefficient**: $r = 0.99$.

Table 7: Performance and fairness comparison during progressive debiasing of sensitive attributes for DistilBERT-base and BERT-base.

| Metric | DistilBERT-base | | | BERT-base | | |
|---|---|---|---|---|---|---|
| | ERM | FLoRa Afr. | FLoRa Fe. | ERM | FLoRa Afr. | FLoRa Fe. |
| Other TPR | 75.2 ± 2.0 | 75.1 ± 1.9 | 77.0 ± 1.7 | 77.1 ± 1.7 | 77.0 ± 1.8 | 78.2 ± 1.6 |
| Afr. TPR | 90.1 ± 1.7 | 90.1 ± 1.7 | 90.1 ± 1.5 | 90.1 ± 1.1 | 85.4 ± 2.0 | 85.4 ± 2.2 |
| Other FPR | 18.9 ± 3.1 | 18.7 ± 2.7 | 19.6 ± 2.5 | 20.5 ± 4.0 | 19.3 ± 3.7 | 20.9 ± 4.1 |
| Afr. FPR | 57.1 ± 2.7 | 52.4 ± 2.9 | 52.4 ± 2.0 | 47.6 ± 4.4 | 33.3 ± 3.6 | 33.3 ± 2.9 |
| DP (R)↓ | 38.2 ± 1.4 | 33.7 ± 1.4 | 32.8 ± 1.1 | 27.1 ± 0.9 | 14.0 ± 1.0 | 12.4 ± 0.7 |
| EOp (R)↓ | 14.9 ± 1.1 | 14.2 ± 1.0 | 13.1 ± 1.0 | 13.0 ± 0.8 | 8.4 ± 1.1 | 7.2 ± 1.0 |
| **EOD (R)↓** | 26.5 ± 0.7 | 24.4 ± 0.6 | **23.0 ± 0.6** | 20.1 ± 0.4 | 11.2 ± 0.6 | **9.8 ± 0.5** |
| Male TPR | 80.5 ± 2.1 | 80.5 ± 2.1 | 80.6 ± 2.0 | 82.5 ± 2.1 | 81.0 ± 1.8 | 81.0 ± 1.7 |
| Fe. TPR | 67.5 ± 2.9 | 67.5 ± 3.1 | 78.6 ± 2.8 | 64.3 ± 2.0 | 64.3 ± 2.0 | 74.2 ± 1.8 |
| Male FPR | 19.7 ± 2.1 | 19.3 ± 1.7 | 20.1 ± 2.1 | 21.0 ± 3.8 | 20.1 ± 3.6 | 21.0 ± 3.7 |
| Fe. FPR | 28.6 ± 3.0 | 28.6 ± 3.1 | 33.0 ± 2.9 | 28.6 ± 2.6 | 28.6 ± 2.6 | 28.6 ± 2.1 |
| DP (G)↓ | 7.4 ± 1.3 | 7.6 ± 1.1 | 12.9 ± 2.2 | 7.6 ± 1.5 | 8.5 ± 1.4 | 7.6 ± 1.0 |
| EOp (G)↓ | 13.0 ± 0.5 | 13.0 ± 0.5 | 2.0 ± 2.1 | 18.2 ± 0.8 | 16.7 ± 0.4 | 8.8 ± 1.4 |
| **EOD (G)↓** | 11.3 ± 1.1 | 11.2 ± 0.7 | **7.4 ± 0.6** | 12.9 ± 1.1 | 12.6 ± 0.9 | **8.2 ± 0.4** |
| **ACC↑** | 79.5 ± 0.2 | 79.6 ± 0.2 | **79.7 ± 0.3** | **79.8 ± 0.3** | 79.6 ± 0.5 | 79.7 ± 0.4 |

* Bold values indicate the best performance in each category, while underlined values represent the second-best results. "R" refers to Race, and "G" refers to Gender.

## 5. Summary

The high correlation coefficients (0.97 for DistilBERT and 0.99 for BERT) indicate a strong positive relationship between the changes in metrics across debiasing stages, demonstrating that mitigating a new bias does not adversely affect previously achieved improvements, effectively preventing catastrophic forgetting.

### C.3 EXPLORING THE IMPACT OF PROCESSING ORDER ON MULTI-SENSITIVE ATTRIBUTES

Table 8: Performance and fairness comparison during progressive debiasing of sensitive attributes for DistilBERT-base and BERT-base.

| Metric | DistilBERT-base | | | BERT-base | | |
|---|---|---|---|---|---|---|
| | ERM | FLoRa Fe. | FLoRa Afr. | ERM | FLoRa Fe. | FLoRa Afr. |
| DP (R)↓ | 38.2 ± 1.4 | 37.8 ± 1.2 | 32.9 ± 1.2 | 27.1 ± 0.9 | 26.7 ± 0.9 | 12.1 ± 1.0 |
| EOp (R)↓ | 14.9 ± 1.1 | 14.7 ± 1.1 | 13.2 ± 1.1 | 13.0 ± 0.8 | 12.4 ± 1.2 | 7.0 ± 1.1 |
| **EOD(R)↓** | 26.5 ± 0.7 | 26.0 ± 0.7 | **23.1 ± 0.7** | 20.1 ± 0.4 | 19.7 ± 0.5 | **9.6 ± 0.7** |
| DP (G)↓ | 7.4 ± 1.3 | 13.0 ± 2.1 | 12.8 ± 2.2 | 7.6 ± 1.5 | 8.0 ± 1.2 | 8.5 ± 1.0 |
| EOp (G)↓ | 13.0 ± 0.5 | 5.0 ± 1.9 | 3.7 ± 1.7 | 18.2 ± 0.8 | 8.9 ± 1.5 | 8.8 ± 1.2 |
| **EOD(G)↓** | 11.3 ± 1.1 | 7.3 ± 0.7 | **7.2 ± 0.6** | 12.9 ± 1.1 | 8.4 ± 0.7 | **8.3 ± 0.5** |
| **ACC↑** | 79.5 ± 0.2 | **79.6 ± 0.3** | 79.6 ± 0.2 | 79.8 ± 0.3 | 79.8 ± 0.5 | **79.9 ± 0.4** |

* Bold values indicate the best performance in each category, while underlined values represent the second-best results. "R" refers to Race, and "G" refers to Gender.

We conducted additional experiments to investigate the impact of varying the sequence of debiasing (FairLORA Race first) and addressing multiple biases simultaneously. As shown in Table 8, the results indicate that the order of debiasing has negligible impact on the final outcomes. This finding aligns with our theoretical explanation that FairLoRA exhibits a "forgetting-avoidance" property, whereby corrections for distinct sensitive attributes are encapsulated in independent LoRA modules.

This design ensures that adjustments made for one attribute do not interfere with those made for others.

Moreover, as illustrated in Table 9, the results demonstrate that whether biases are mitigated sequentially or simultaneously, the overall outcomes remain largely consistent. This robustness arises from FairLoRA's modular architecture, which stores adjustments for each sensitive attribute in separate LoRA modules, allowing independent corrections without cross-attribute interference.

Table 9: Comparison of Progressive Debiasing and Simultaneous Debiasing Approaches.

| Metric | DistilBERT-base | | | BERT-base | | |
|---|---|---|---|---|---|---|
| | Afr.Fisrt | Fe.Fisrt | Together | Afr.Fisrt | Fe.Fisrt | Together |
| DP (R)↓ | 32.8 ± 1.1 | 32.9 ± 1.2 | 33.2 ± 1.2 | 12.4 ± 0.7 | 12.1 ± 1.0 | 12.8 ± 1.1 |
| EOp (R)↓ | 13.1 ± 1.0 | 13.2 ± 1.1 | 13.3 ± 1.1 | 7.2 ± 1.0 | 7.0 ± 1.1 | 7.5 ± 1.2 |
| **EOD(R)↓** | **23.0 ± 0.6** | 23.1 ± 0.7 | 23.3 ± 0.8 | 9.8 ± 0.5 | **9.6 ± 0.7** | 10.0 ± 0.7 |
| DP (G)↓ | 12.9 ± 2.2 | 12.8 ± 2.2 | 13.1 ± 2.3 | 7.6 ± 1.0 | 8.5 ± 1.0 | 8.0 ± 1.2 |
| EOp (G)↓ | 2.0 ± 2.1 | 3.7 ± 1.7 | 4.7 ± 2.2 | 8.8 ± 1.4 | 8.8 ± 1.2 | 9.0 ± 1.4 |
| **EOD(G)↓** | 7.4 ± 0.6 | **7.2 ± 0.6** | 7.4 ± 0.8 | **8.2 ± 0.4** | 8.3 ± 0.5 | 8.5 ± 0.7 |
| **ACC↑** | **79.7 ± 0.3** | 79.6 ± 0.2 | 79.6 ± 0.3 | 79.7 ± 0.4 | **79.9 ± 0.4** | 79.7 ± 0.4 |

[*] Afr.First refers to applying FairLoRA to address bias for African Americans first, while Fe.First refers to addressing bias for females first, and Together represents simultaneous bias mitigation for both groups.

## D    IMPACT OF THRESHOLD ON DISCRIMINATOR TPR AND FPR FOR DEMOGRAPHIC GROUPS

**African American Group Analysis (Left Pair of Plots in Figure 3)** The top-left plot illustrates the variation of True Positive Rate (TPR) and False Positive Rate (FPR) for the "African American" group as a function of the threshold. As the threshold increases, both TPR and FPR decrease. The reduction in TPR suggests that a higher threshold leads to stricter classification, reducing the number of true positives. Meanwhile, the rapid decrease in FPR indicates fewer false positives.

The bottom-left plot shows the TPR/FPR ratio across different thresholds. This ratio peaks at approximately 0.7-0.8, indicating an optimal balance between TPR and FPR. Beyond this peak, the ratio declines, suggesting diminishing benefits from further increasing the threshold due to a disproportionate reduction in TPR compared to the decline in FPR. Therefore, this peak threshold can be used to guide optimal threshold selection, ensuring fairness and maintaining model performance.

**Female Group Analysis (Right Pair of Plots in Figure 3)** The top-right plot shows the changes in TPR and FPR for the "Female" group, following a similar pattern to the "African American" group. As the threshold increases, both TPR and FPR decrease, with higher thresholds making the model stricter, leading to a reduction in both true positives and false positives.

The bottom-right plot depicts the TPR/FPR ratio, which also peaks around the 0.7-0.8 threshold range, indicating the threshold range that maximizes classification efficiency for the "Female" group. After this peak, the ratio starts to decline, suggesting that further increases in the threshold reduce classification effectiveness. Thus, selecting a threshold near this peak ensures optimal fairness while retaining classification accuracy.

**Summary** For both the "African American" and "Female" groups in the HateXplain dataset, the TPR/FPR ratio reaches its peak around a threshold of 0.7-0.8, indicating that this range provides the optimal balance between fairness and classification performance. For other datasets, a similar analysis can be conducted to determine the optimal threshold range that ensures FairLoRA effectively mitigates biases while maintaining overall model efficacy.

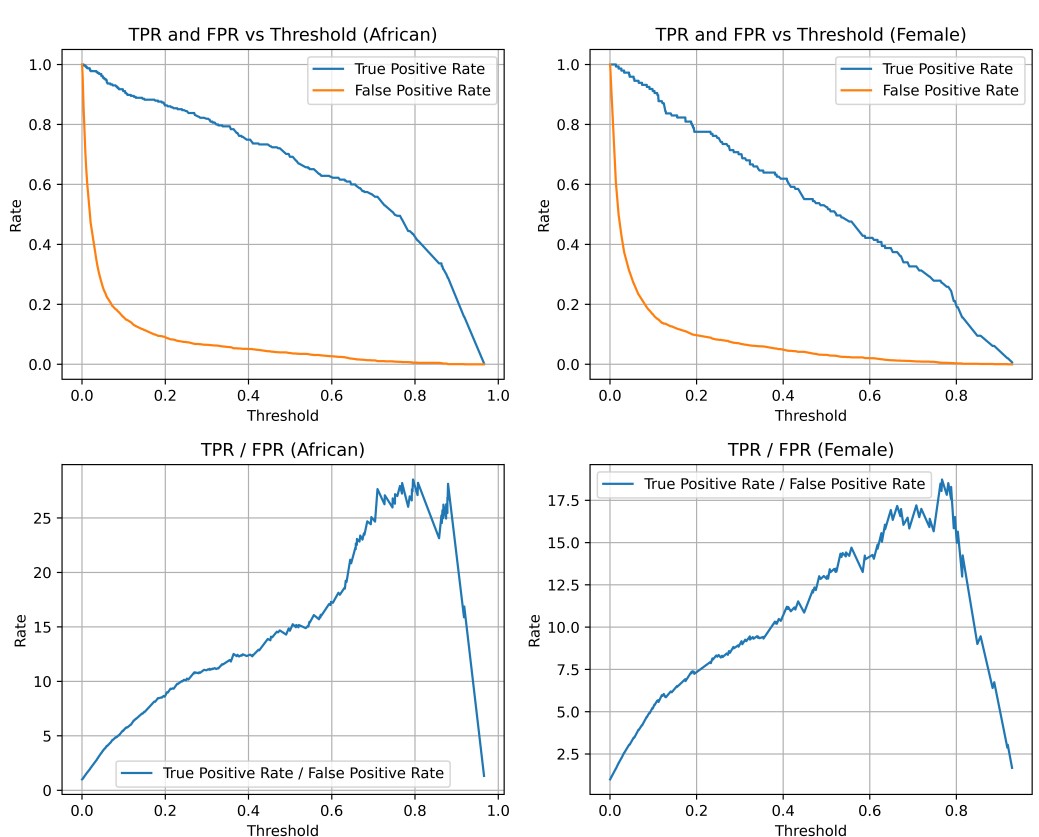

Figure 3: TPR and FPR Analysis with TPR/FPR Ratio for African American and Female Groups across Different Thresholds.

