# OpenReview forum: "FairLoRA: Targeted Bias Mitigation without Performance Loss"
_ICLR.cc/2025/Conference — Submitted to ICLR 2025_

### Official Review · Reviewer_9eQv · 2024-11-01

**Soundness:** 2
**Presentation:** 2
**Contribution:** 1
**Rating:** 3
**Confidence:** 4

**Summary:**

Mitigating bias becomes essential to ensure trustworthy AI systems, but it is often not effective and requires extensive bias supervisions. The authors proposed FairLoRA, which incorporates LoRA technique to mitigate bias while preserving the performance, and a bias discriminator to reduce the reliance on bias supervision. Theoretical analysis presents the condition when bias mitigation without performance loss is effective.

**Strengths:**

- Experimental results on three standard datasets are provided.
- It extends FairLoRA framework to address multiple bias attributes

**Weaknesses:**

- What is the meanings of theoretical analysis? It does not guarantee that FairLoRA algorithm ensures fair results (for meaningful theoretical analysis, at least it needs to show that FairLoRA satisfy Eq.(7)).
- Hard to find the novelty and motivation of the proposed method. Integrating discriminators or adopting LoRA for fairness are well investigated in this literature [1,2,3,4].
- It still requires the knowledge of the presence of sensitive attribute, which could be not applicable to unsupervised scenarios.
- Writing should be improved. In Section 3, it is hard to follow to understand the overall training pipeline. (Why the details and algorithms are in Appendix?) The caption of Figure 1 is also not well described.
- Performance improvements are not strong and experimental results are not comprehensive. Fairness-accuracy trade-off is a well-known phenomena in this literature, and the results merely present the trade-off. It is hard to verify whether the proposed method actually improves the fairness. Most of improvements are within the standard deviations.

[1] On Fairness of Low-Rank Adaptation of Large Models, Ding et al., arxiv 2024

[2] Finetuning Text-to-Image Diffusion Models for Fairness, Shen et al., ICLR 2024

[3] A Trip Towards Fairness: Bias and De-Biasing in Large Language Models, ACL 2023

[4] Low-rank finetuning for LLMs: A fairness perspective, Das et al., arxiv 2024

**Questions:**

- What if the sequence of debiasing is changed (gender → racial) or eliminating multiple bias simultaneously. I think it could adopt multiple discriminators at once.
- No worst-group accuracy results in Table 2.
- Limitations of this work should be addressed.

---

> ### Author Response · Authors · 2024-11-25
>
> ## **Response to Question 1:**
>
> Thank you for your thoughtful suggestion. We conducted additional experiments to investigate the impact of changing the debiasing sequence (e.g., gender → racial) and eliminating multiple biases simultaneously.
>
> The results, as shown in **Table 8**, demonstrate that altering the order of debiasing has minimal effect on the final outcomes. This aligns with our explanation in the paper that FairLoRA possesses a "forgetting-avoidance" property, where corrections for different sensitive attributes are stored in separate LoRA modules, ensuring that adjustments to one attribute do not interfere with others.
>
> Furthermore, **Table 9** illustrates that whether biases are mitigated sequentially or simultaneously, the final results remain nearly identical. This is due to the fact that we store corrections for each bias in separate LoRA modules, which allows for independent adjustments to each sensitive attribute without negatively impacting the others.
>
> Additionally, we focused on validating FairLoRA's ability to incrementally mitigate biases because, as society evolves, new biases will inevitably emerge. If each newly discovered bias requires retraining the entire model, it would result in significant costs and delays. Therefore, we believe that the ability to incrementally mitigate biases is essential for practical deployment in real-world scenarios.
>
> ### Table 8: Performance and fairness comparison during progressive debiasing of sensitive attributes for DistilBERT-base and BERT-base
>
> | **Metric**       | **DistilBERT-base** |           |           | **BERT-base**      |           |           |
> |-------------------|---------------------|-----------|-----------|--------------------|-----------|-----------|
> |                   | **ERM**            | **FLoRa Fe.** | **FLoRa Afr.** | **ERM**           | **FLoRa Fe.** | **FLoRa Afr.** |
> | **DP (R) ↓**     | 38.2 ± 1.4         | 37.8 ± 1.2 | 32.9 ± 1.2 | 27.1 ± 0.9        | 26.7 ± 0.9 | 12.1 ± 1.0 |
> | **EOp (R) ↓**    | 14.9 ± 1.1         | 14.7 ± 1.1 | 13.2 ± 1.1 | 13.0 ± 0.8        | 12.4 ± 1.2 | 7.0 ± 1.1  |
> | **EOD (R) ↓**    | 26.5 ± 0.7         | *26.0 ± 0.7* | **23.1 ± 0.7** | 20.1 ± 0.4        | *19.7 ± 0.5* | **9.6 ± 0.7** |
> | **DP (G) ↓**     | 7.4 ± 1.3          | 13.0 ± 2.1 | 12.8 ± 2.2 | 7.6 ± 1.5         | 8.0 ± 1.2  | 8.5 ± 1.0  |
> | **EOp (G) ↓**    | 13.0 ± 0.5         | 5.0 ± 1.9  | 3.7 ± 1.7  | 18.2 ± 0.8        | 8.9 ± 1.5  | 8.8 ± 1.2  |
> | **EOD (G) ↓**    | 11.3 ± 1.1         | *7.3 ± 0.7* | **7.2 ± 0.6** | 12.9 ± 1.1        | *8.4 ± 0.7* | **8.3 ± 0.5** |
> | **ACC ↑**        | 79.5 ± 0.2         | **79.6 ± 0.3** | **79.6 ± 0.2** | *79.8 ± 0.3*      | *79.8 ± 0.5* | **79.9 ± 0.4** |
>
> ---
>
> ### Table 9: Comparison of Progressive Debiasing and Simultaneous Debiasing Approaches
>
> | **Metric**       | **DistilBERT-base**         |                 |             | **BERT-base**            |                 |             |
> |-------------------|----------------------------|-----------------|-------------|--------------------------|-----------------|-------------|
> |                   | **Afr. First**            | **Fe. First**   | **Together** | **Afr. First**          | **Fe. First**   | **Together** |
> | **DP (R) ↓**     | 32.8 ± 1.1                 | 32.9 ± 1.2      | 33.2 ± 1.2  | 12.4 ± 0.7              | 12.1 ± 1.0      | 12.8 ± 1.1  |
> | **EOp (R) ↓**    | 13.1 ± 1.0                 | 13.2 ± 1.1      | 13.3 ± 1.1  | 7.2 ± 1.0               | 7.0 ± 1.1       | 7.5 ± 1.2   |
> | **EOD (R) ↓**    | **23.0 ± 0.6**             | *23.1 ± 0.7*    | 23.3 ± 0.8  | *9.8 ± 0.5*             | **9.6 ± 0.7**   | 10.0 ± 0.7  |
> | **DP (G) ↓**     | 12.9 ± 2.2                 | 12.8 ± 2.2      | 13.1 ± 2.3  | 7.6 ± 1.0               | 8.5 ± 1.0       | 8.0 ± 1.2   |
> | **EOp (G) ↓**    | 2.0 ± 2.1                  | 3.7 ± 1.7       | 4.7 ± 2.2   | 8.8 ± 1.4               | 8.8 ± 1.2       | 9.0 ± 1.4   |
> | **EOD (G) ↓**    | *7.4 ± 0.6*                | **7.2 ± 0.6**   | *7.4 ± 0.8* | **8.2 ± 0.4**           | *8.3 ± 0.5*     | 8.5 ± 0.7   |
> | **ACC ↑**        | **79.7 ± 0.3**             | *79.6 ± 0.2*    | *79.6 ± 0.3* | *79.7 ± 0.4*            | **79.9 ± 0.4**  | *79.7 ± 0.4* |

---

> ### Author Response · Authors · 2024-11-25
>
> ---
>
> ## **Response to Question 2:**
>
> Thanks for your question. In scenarios involving two sensitive attributes (Race and Sex), the worst-group is typically defined as the group with the lowest performance among all possible combinations of sensitive attributes and labels. In our experiments, this results in a total of 8 groups (2 Race attributes × 2 Sex attributes × 2 labels).
>
> In multi-attribute scenarios, WGA primarily reflects the bias of the most underrepresented group, which often masks the nuanced effects of bias mitigation for individual attributes. For example, when mitigating gender bias, the WGA might still be dominated by the smallest group defined by race, thereby overshadowing FairLoRA's specific contributions to reducing gender bias.
>
> This limitation makes WGA less effective at demonstrating the stepwise mitigation of biases across multiple sensitive attributes, which is a key focus of our framework. Thus, we did not include changes in WGA scores.
>
> ---
>
> ## **Response to Weaknesses 1:**
>
> We apologize for any lack of clarity in our initial presentation. In the revised version, we have reorganized the theoretical analysis section to better explain its purpose and conclusions.
>
> The theoretical analysis aims to derive the conditions under which the FairLoRA framework can fine-tune a model without compromising the original model's performance. This analysis is specifically designed for the FairLoRA framework, where TPR (True Positive Rate) and FPR (False Positive Rate) represent the performance of the discriminator in FairLoRA, \(M\) denotes the original model, and \(M_{\text{lora}}\) represents the model after adding the LoRA module.
>
> Our results show that when the discriminator in FairLoRA satisfies Eq.(7), the framework ensures that fine-tuning does not degrade the original model's performance. Moreover, we provide an approximate condition:$\frac{\text{TPR}}{\text{FPR}} \geq \frac{1-p}{p}$,
> where \(p\) represents the proportion of sensitive labels. We also explain why this approximation is reasonable in most practical scenarios. This approximation offers a practical and interpretable way to assess the framework's ability to maintain performance during fine-tuning.
>
> We hope this clarification addresses your concerns and demonstrates the theoretical grounding of our method.
>
> ---
>
> ## **Response to Weaknesses 2:**
>
> While prior works have explored fine-tuning using LoRA and adversarial learning with discriminators, our proposed framework differs significantly in its design and objectives. Specifically, our approach integrates a pre-trained discriminator capable of identifying sensitive attributes directly into the model architecture. Based on the discriminator's outputs, we selectively activate LoRA modules that are specifically trained to mitigate biases related to sensitive attributes.
>
> The primary goal of our method is to enhance the model's fairness without compromising its original performance. To this end, we provide a theoretical guarantee tailored specifically to the FairLoRA framework. This theoretical guarantee ensures that models fine-tuned with FairLoRA can improve fairness under the condition of maintaining the original model's performance. Importantly, this guarantee is unique to the FairLoRA structure and does not generalize to other frameworks or architectures.
>
> We believe this combination of selective LoRA activation and a performance-preserving theoretical foundation distinguishes FairLoRA from existing methods in the literature.
>
> ---
>
> ## **Response to Weaknesses 3:**
>
> Our method does require access to some labeled data. However, we believe that both labeled and unsupervised bias mitigation approaches have distinct advantages and application scenarios.
>
> Labeled bias mitigation allows for precise identification of data or behaviors that contain biases, enabling a more targeted and accurate debiasing process while minimizing the risk of misclassification or omission. This level of precision is particularly valuable in scenarios where sensitive attributes are explicitly available or critical for the application context.
>
> Since labeled and unsupervised bias mitigation methods serve different purposes and are suited to different domains, we argue that focusing on labeled bias mitigation should not be viewed as a weakness. Instead, it represents a deliberate design choice to address specific challenges that require a more controlled and interpretable debiasing process.

---

> ### Author Response · Authors · 2024-11-25
>
> ## **Response to Weaknesses 4:**
>
> Thank you for pointing this out. We apologize for the lack of clarity in our initial submission. We have revised the relevant sections to improve readability and ensure the training pipeline is easier to follow.
>
> The details and algorithms were placed in the Appendix due to page limitations. However, in the revised version, we have restructured the content to provide better context and flow in Section 3, while keeping key explanations concise and accessible. Additionally, we have updated the caption of Figure 1 to provide a more comprehensive description.
>
> We hope these revisions address your concerns and improve the overall clarity of the manuscript.
>
> ---
>
> ## **Response to Weaknesses 5:**
>
> Thank you for your valuable feedback. The fairness-accuracy trade-off is indeed a well-recognized phenomenon in this field. However, FairLoRA is specifically designed to improve fairness without sacrificing the model's original performance, distinguishing it from methods that rely on such trade-offs.
>
> Because FairLoRA avoids trading accuracy for fairness, the fairness improvements may appear more limited compared to approaches that explicitly compromise performance. Nevertheless, our method ensures these improvements through both theoretical guarantees and experimental validation. From a theoretical perspective, we demonstrate that FairLoRA maintains accuracy while enhancing fairness under defined conditions. Experimentally, the results consistently show improvements in fairness metrics (e.g., WGA and EOD) while preserving accuracy.
>
> Additionally, Experiment 2 (Table 2) demonstrates the incremental debiasing capability of FairLoRA. By storing information for different biases in separate modules, FairLoRA avoids catastrophic forgetting and achieves composability, enabling the effective mitigation of multiple biases over time.
>
> While the fairness improvements may seem modest, they are significant in practical scenarios where maintaining performance is critical. We will further expand experiments in future work to comprehensively evaluate the method's potential across a broader range of datasets and metrics. We hope this explanation clarifies the novelty and effectiveness of FairLoRA in addressing the fairness-accuracy trade-off.

---

> > ### Comment · Reviewer_9eQv · 2024-12-01
> >
> > Thank you for the detailed response. Some of my concerns have been addressed, but I still find some aspects challenging to follow. Specifically, I still think the theoretical analysis does not guarantee FairLoRA's fairness, but rather outlines the conditions under which the overall performance is not compromised. For example, why FairLoRA is particularly advantageous for achieving Eq. (7) compared to other post-processing debiasing methods like DFR? I find it difficult to connect the analysis and the proposed framework design.

---

> > > ### Author Response · Authors · 2024-12-01
> > >
> > > Thank you for your thoughtful feedback and for raising these important points. We understand your concerns and would like to clarify the theoretical analysis and its connection to the proposed FairLoRA framework.
> > >
> > >   ---
> > >
> > >   You are correct that our theoretical analysis primarily focuses on verifying the conditions under which performance trade-offs can be avoided. It is widely acknowledged that fine-tuning a model on balanced datasets or applying other fine-tuning techniques generally enhances fairness. Therefore, we did not aim to prove this phenomenon itself. Instead, our objective was to explore whether fairness improvements can be achieved without compromising the model's original performance and to provide conditions under which this trade-off can be avoided.
> > >
> > >   Performance trade-offs are a common occurrence in existing models and methods due to the inherent nature of model parameter adjustments. When fine-tuning is performed to enhance one capability, it often results in the deterioration of others because parameter updates alter the model's overall behavior. **Our goal was to propose a method to address this issue—improving fairness while maintaining the original performance of the model**.
> > > ﻿
> > >   To achieve this, we introduced the FairLoRA framework and demonstrated its feasibility in avoiding trade-offs through both theoretical and experimental analyses. **Specifically, Eq. (7) is unique to FairLoRA, as it relates to the TPR and FPR of the discriminator in FairLoRA, rather than the overall model**. We have clarified this distinction in the revised manuscript. **Only FairLoRA incorporates discriminator modules that selectively apply LoRA fine-tuning to underrepresented groups, a structure absent in comparison methods like DFR. As a result, these methods cannot satisfy Eq. (7) as FairLoRA does**.
> > > ﻿
> > >   Our comparison with post-processing methods like DFR is intended to highlight the versatility of the FairLoRA approach. Regardless of whether the model is trained using ERM or other fairness-promoting algorithms (including post-processing methods), FairLoRA can further improve fairness without compromising the original performance. This distinguishes FairLoRA as a flexible and effective approach for enhancing fairness across a range of scenarios.
> > > ﻿
> > >   We hope this explanation clarifies the relationship between our theoretical analysis and the design of FairLoRA. If you have additional questions or require further details, we would be happy to provide further clarification.

---

### Official Review · Reviewer_dZWF · 2024-11-01

**Soundness:** 2
**Presentation:** 1
**Contribution:** 2
**Rating:** 3
**Confidence:** 4

**Summary:**

The paper presents a framework that given a pre-trained network, finetunes a different set of LoRA weights for each sensitive attribute group and the LoRA weights are applied to the pre-trained weights for each sample if a `Discriminator` network is activated. The Discriminator network is trained using a fairness-oriented objective function.

**Strengths:**

- The paper uses multiple fairness metrics for evaluating their method.
- They further provides theorems and proves for their theorems.

**Weaknesses:**

- Details about the models and experiments done is vague.
- Details of the models used for Experiment 1 and 3 are not mentioned. Also, none of the models are "state-of-the-art" as claimed in the paper.
- No exact mentioning of which splits of which datasets were used for training and testing and the claim "FairLoRA operates under a partial-information setting, where group labels are observed only for the validation set, not for the (much larger) training set." is incorrect based on the experiments explained. It can "also" be applied to the partial-information setting.
- The method is never "compared" one-to-one with another method. All tables merely mention the results after their method *is added* to ERM and previous fairness frameworks and improves them. But their method is *not* compared to either of them head-to-head to my understanding.
- Limited hyper parameter tuning and using a single set of hyper params for all models. (or the complete set was not reported) Also, number of seeds were not reported.
- Given the standard deviations, the differences in Table 1 don't seem significant for in most comparisons.

**Questions:**

- Please address the weaknesses mentioned above.
- What is the model used in the experiments of Section 5.2 for each of ERM, GroupDRO, DFR, and Lu et al? Also, are they all the same models?
- Do you have insights as to why the EoD(R) improves in the final stage of experiment 2 in Table 2? I.e., why does the EoD(R) further improve when mitigating gender bias?

---

> ### Author Response · Authors · 2024-11-25
>
> ## **Response to Weaknesses 1&2:**
>
> We apologize for the lack of clarity in our original manuscript. We have revised the relevant sections to provide the requested details.  Specifically, for Experiments 1 and 3, the model used for CelebA is a 8-layer Vision Transformer (ViT) model. For the MultiNLI dataset, we used the BERT-base model. These choices of datasets and models align with those used in the methods we compared against in our experiments.
>
> Regarding the claim of ``state-of-the-art'' models, we acknowledge that our description may have been overly general. The methods we compared against are well-established and widely recognized in the field, but may not strictly qualify as "state-of-the-art'' in the current sense. We have revised the manuscript to remove this claim to ensure rigor.
>
> ---
>
> ## **Response to Weaknesses 3:**
>
> Thank you for your valuable feedback. We apologize for the lack of clarity regarding the dataset splits.
>
> - For CelebA and MultiNLI, we used the official splits provided in the respective documentation, following the standard training and test set divisions.
> - For HateXplain, since the official split is not provided, we followed the approach of Lu et al., where 50% of the samples were used as the test set.
>
> Regarding the claim about the partial-information setting, we acknowledge that this description was not entirely accurate. We have revised this section to clarify that "FairLoRA operates under a partial-information setting, where group labels are observed only for a subset of the training set." Thank you again for pointing this out, and we hope the revision better reflects the experimental setup.
>
> ---
>
> ## **Response to Weaknesses 4:**
>
> Thank you for raising this point. Below, we clarify FairLoRA's design objectives and provide the rationale for our evaluation approach:
>
> 1. **FairLoRA's Design Objectives and Unique Contribution**
> FairLoRA is designed as a *post-processing framework* to enhance fairness while preserving the original model performance.  Unlike methods that optimize fairness by sacrificing accuracy or other metrics, FairLoRA prioritizes a **trade-off-free improvement**, ensuring no degradation in performance. This fundamental design goal differentiates FairLoRA from other methods and positions it as a complementary, rather than competing, approach.
>
> 2. **Rationale for Experimental Design**
>    - **Compatibility and Generalizability**:  FairLoRA's strength lies in its compatibility with and enhancement of existing fairness methods, such as ERM, GroupDRO, and DFR.  Therefore, instead of conducting one-to-one comparisons, we focused on demonstrating its **synergistic effects** when integrated with these methods. This approach highlights FairLoRA’s **broad applicability and composability**, which are central to its contribution.
>
>    - **Avoiding Misleading Comparisons**:  FairLoRA does not aim to maximize fairness improvement at any cost, as some methods designed for specific fairness metrics do. Direct one-to-one comparisons with such methods may misrepresent its value, as FairLoRA emphasizes balanced improvements across fairness and performance without sacrificing one for the other.
>
> 3. **FairLoRA's Practical Value**
> The experimental results are structured to emphasize FairLoRA's **real-world practicality**: its ability to enhance fairness across diverse methods without compromising baseline performance metrics.
> This trade-off-free and general-purpose nature addresses common limitations in fairness optimization and showcases its unique contribution.
>
> We appreciate the suggestion and are open to incorporating additional head-to-head comparisons in future work to further contextualize its performance relative to other methods.
>
> ---
>
> ## **Response to Weaknesses 5:**
>
> Thank you for your valuable comments. The primary objective of our work is to validate the ability of FairLoRA to operate without requiring a trade-off between performance and fairness. As such, the specific choice of hyperparameters or models does not impact the core validation of our method.
>
> However, we acknowledge the importance of providing full details for reproducibility. In the revised manuscript, we have added this information in the supplementary materials. Specifically, we report the random seeds used in our experiments, which are seed 5, 15, 25, 35, and 45. For all models, we used the default hyperparameter settings provided in the original research papers, as no further tuning was conducted.
>
> We hope this clarifies the experimental setup, and we appreciate your suggestion for greater transparency.

---

> ### Author Response · Authors · 2024-11-25
>
> ## **Response to Weaknesses 6:**
>
> Thank you for your insightful comment. We apologize if the presentation in Table 1 was unclear, leading to potential misunderstandings. Our FairLoRA Minority method is designed to fine-tune the model with the goal of improving fairness (measured by WGA and EOD) without sacrificing original performance (specifically, accuracy).
> In other words, we aim to enhance fairness while ensuring that ACC does not decrease.
>
> In contrast, methods like ERM focus primarily on optimizing accuracy, which results in higher fairness metrics. When FairLoRA Minority is applied, ACC remains unchanged, while WGA and EOD show significant improvements. For instance, on the CelebA and MultiNLI datasets, WGA improved from 77.9 to 82.0 and from 67.3 to 71.0, respectively.
>
> For fairness-oriented methods such as GroupDRO, DFR, and Lu et al., since these methods have already achieved relatively high fairness, the improvements from applying FairLoRA Minority are smaller. However, there is still a noticeable increase in fairness. These fairness-centric methods typically do not achieve the highest accuracy because they prioritize fairness over performance. On the other hand, applying FairLoRA Majority allows for further accuracy improvements without sacrificing fairness significantly.
>
> Our method does not require a trade-off between fairness and performance, which is why the fairness gains are not as dramatic as those seen in methods that explicitly sacrifice performance to improve fairness.
>
> ---
>
> ## **Response to Question 1:**
>
> Responses to all weaknesses have been provided in the sections above.
>
> ---
>
> ## **Response to Question 2:**
>
> - Thank you for your question. We apologize for any potential confusion in the original manuscript. To clarify, in the experiments involving the same datasets, the models used for ERM, GroupDRO, DFR, and Lu et al.'s method are indeed the same. The differences among these methods lie in the fine-tuning strategies applied to the pre-trained models.
>
> - Specifically, for the CelebA dataset, we utilized a 8-layer Vision Transformer (ViT) model. For the MultiNLI dataset, we employed the BERT-base model. These choices of datasets and models are consistent with those used in the methods we compared against in our experiments. In the revised manuscript, we have made further clarifications to ensure that the description is clear and avoids any ambiguity. We hope this resolves your concern, and we appreciate your attention to this detail.
>
> ---
>
> ## **Response to Question 3:**
> Thank you for your thoughtful question. The improvement in EoD(R) during the final stage of Experiment 2 can be attributed to the complex interplay between Race and Sex biases, which are not entirely independent of each other. These two sources of bias can interact in ways that amplify or mitigate each other's effects in the model.
>
> For instance, certain data points may exhibit biased features related to both race and sex simultaneously. By mitigating one type of bias (e.g., gender bias), we may also reduce some of the associated bias in the other attribute (e.g., race bias). Specifically, certain race-gender combinations (e.g., a particular racial and gender group) may be subject to distinct forms of biased treatment. Eliminating gender bias could therefore help alleviate some of the racial bias present in these specific combinations.
>
> We hope this insight helps clarify the observed improvement in EoD(R). We have made this reasoning more explicit in the revised manuscript for better clarity.

---

> > ### Comment · Reviewer_dZWF · 2024-12-02
> > **Response**
> >
> > Dear authors,
> >
> > Thank you for providing additional explanations, clarifications, and for revising the manuscript. I appreciate the effort you’ve put into addressing the concerns raised.
> >
> > From your explanations, it seems FairLoRA is intended as a post-processing framework. However, given that most improvements in fairness metrics fall within the standard deviations—such as the WGA metric on CelebA and MultiNLI—it remains challenging to assess the clear impact of FairLoRA.
> >
> > Regarding the relationship between gender and race as separate features in the data, I found the explanation provided to be somewhat unconvincing. However, I recognize that this could reflect observations from the experiments, and I encourage the authors to explore and present additional explanations if possible.
> >
> > After reviewing the rebuttal and revised paper, I acknowledge the significant effort made, but I remain unconvinced of the effectiveness of FairLoRA to the extent claimed. As such, I plan to maintain my initial score.

---

> > > ### Author Response · Authors · 2024-12-02
> > >
> > > We regret that our method did not fully meet your expectations, despite its theoretical and experimental validation in effectively addressing the trade-off issue under broad and practical conditions.
> > >
> > > Our approach and underlying formulas are highly generalizable, and we have already demonstrated its robustness in avoiding trade-offs across additional tasks beyond those presented in this work. While it has not yet gained your full recognition, we remain confident that our method will prove its value as it is applied to a wider range of scenarios in the future.
> > >
> > > We respect your decision to maintain the initial score and sincerely thank you for acknowledging the effort we have invested in addressing the concerns raised. Thank you once again for your time and thoughtful feedback!

---

### Official Review · Reviewer_7Nfm · 2024-11-03

**Soundness:** 3
**Presentation:** 3
**Contribution:** 3
**Rating:** 8
**Confidence:** 3

**Summary:**

This paper proposes a technique that boosts the model performance on minority groups, by (1) training a cascade of discriminators that predict whether or not samples contain belong to a minority group and (2) selectively applying LoRA updates on predicted minority samples.

**Strengths:**

## Strengths:
- There are very few conditions for FairLORA to be applied, making it a versatile solution for enhancing the fairness of pretrained models.
- Significant improvements in terms of WGA are present even when very few sensitive attribute labels are present (Table 3)
- Experiments in both vision and language are quite compelling

**Weaknesses:**

## Weaknesses:
- The theoretical contribution of this work is a bit overstated. The final conclusion of Thm. 3, that the ratio between TPR and FPR must be high in order to prevent performance degradation, seems obvious. Also, Thms 1 and 2 should be lemmas rather than stand alone theorems, since they seem like intermediate steps to build up Thm 3.
- As far as I understand it, one must sequentially (1) adapt the model to the task with standard techniques and (2) apply FairLoRA. However, what if the base adaptation technique learns representations that do not contain much information about the sensitive attribute? This can happen if y does not depend on the sensitive attribute or if the adaptation technique explicitly encourages invariance to the sensitive attribute [1].

## Minor:
- In table 3, WRA -> WGA

## References
- 1. Learning Adversarially Fair and Transferable Representations (https://arxiv.org/pdf/1802.06309)

**Questions:**

See Weaknesses

---

> ### Author Response · Authors · 2024-11-25
>
> ## **Response to Weaknesses 1**:
> Thanks to the reviewer's insightful comments, we have made the following revisions and clarifications in response:
>
> 1. **Regarding Thms 1 and 2:**
>    We fully agree with the reviewer's suggestion that Thms 1 and 2 are intermediate steps in the proof of Thm 3. In the revised manuscript, we have rephrased these as lemmas to more accurately reflect their role in the theoretical development.
>
> 2. **Clarification of Theoretical Contribution:**
>    We acknowledge the reviewer's concern regarding the perceived obviousness of the conclusion in Thm 3, specifically the requirement for a high ratio between TPR and FPR. To avoid any ambiguity, we would like to clarify that the TPR and FPR referred to in Thm 3 are not those associated with the final classification results. Rather, they pertain to the metrics associated with our specifically designed discriminator (unique to the FairLoRA framework), which determines whether the LoRA module should be activated to adjust the model for improved fairness.
>
> More importantly, the core contribution of Thm 3 is the demonstration of the specific conditions under which the FairLoRA framework can improve fairness during fine-tuning without sacrificing model performance. This result breaks away from the conventional trade-off between performance and fairness that is typically observed in existing methods. In other words, when certain conditions are met, the FairLoRA framework allows for a significant improvement in fairness without any degradation in original performance, thus overcoming the typical performance-fairness trade-off seen in prior work.
>
> ---
>
> ## **Response to Weaknesses 2**:
> We appreciate the reviewer's thoughtful question and offer the following clarifications:
>
> 1. **Regarding the presence of sensitive attribute information in model representations:**
>    In most cases, the hidden layers of a pretrained model contain rich token representations that capture a variety of features, including information related to sensitive attributes. We aggregate these representations through an attention pooling mechanism and train a discriminator to classify whether sensitive attribute information is present. If such information is detected, the FairLoRA discriminator effectively activates the LoRA module to adjust the model and reduce bias.
>
> 2. **If the hidden layers do not contain sensitive attribute information:**
>    In situations where the model has learned representations that are invariant to the sensitive attribute—either because the task itself does not depend on it or due to the use of adaptation techniques that explicitly encourage invariance to sensitive attributes—our discriminator will be unable to detect the presence of sensitive attribute information. In this case, we believe that no further intervention from the FairLoRA framework is necessary, as the model has already learned to be unbiased with respect to that attribute. Specifically, when the representations for sensitive and non-sensitive attributes are indistinguishable, no bias exists.
>
> We appreciate the reviewer’s thoughtful comments and insightful suggestions, which have improved the quality of our work.

---

> > ### Comment · Reviewer_7Nfm · 2024-11-30
> >
> > I thank the authors for making the requested changes. Since all of my concerns are addressed, I have updated my score.

---

> > > ### Author Response · Authors · 2024-12-01
> > >
> > > Thank you for your positive feedback and for taking the time to review our work thoroughly! We greatly appreciate your thoughtful comments and are glad that the revisions have addressed your concerns. Your input has been invaluable in improving the quality of our manuscript!

---

### Official Review · Reviewer_ug8P · 2024-11-07

**Soundness:** 3
**Presentation:** 3
**Contribution:** 3
**Rating:** 6
**Confidence:** 3

**Summary:**

This paper introduces FairLoRA, an approach to finetuning for debiasing ML models while preserving model performance. FairLORA approaches debiasing by combining a group discriminator with LoRA modules. It works by activating the LoRA block only if the discriminator determines the sample belongs to a minority group. Otherwise, the model processes the sample through its unmodified layers without LoRA intervention. FairLoRA demonstrates improvements in minority group performance while maintaining the performance for the majority group. Additionally, FairLoRA supports multiple attribute debiasing by performing chained finetuning.

**Strengths:**

- FairLoRA leverages the parameter efficient finetuning method LoRA, meaning it’s a much more efficient technique compared to existing fairness finetuning approaches. Most previous fairness mitigations require either *(i)* retraining the model from scratch on group-balanced data or *(ii)* finetuning a pretrained model. It’s a solid idea to use a modified version of LoRA.
- FairLoRA is also strategic because it selectively activates the LoRA blocks. Only samples from minority groups use the LoRA blocks; the other samples are processed by the unchanged base model. This is one way the model is able to maintain majority-group performance while seeing the minority-group performance gains.
- FairLoRA also supports multiple sensitive attributes. I like this part particularly because often, fairness mitigations are designed in either a single attribute setup. It addresses the component of intersectionality that is not supported in most prior work.
- From the experimental side, many performance metrics are used to better support the claim of improved/maintained performance. I also appreciate that performing chained finetuning does not decrease performance for earlier debiased groups.

**Weaknesses:**

- The largest weakness is that FairLoRA requires a balanced dataset for finetuning. This means that existing datasets have to be downsized because minority groups are nearly always, by definition, much less represented in the data. (In the instances where they’re not less represented, they’re often more stereotypically represented). The lack of balanced labels is one of the main reasons for such fairness disparities in ML models. The other issue I see with requiring balanced labels is that groups that are extremely underrepresented in the data will likely not benefit from FairLoRA. By the time the dataset has been balanced such that size(majority group) == size(minority group), the dataset will be very small.
- The experiments are a bit limited. For instance, for CelebA, the paper tests the bias of “male” and “blond hair”. Because CelebA has so many attributes, at least one other combination should be evaluated as well to better validate FairLoRA.

**Questions:**

**Questions**
- Figure 1’s caption describes the discriminator as essentially being a simple binary classifier, trained to predict whether the sample is from the majority group or minority group. Then in lines 201-205, the discriminator’s objective function appears more detailed: “The discriminator is trained using a fairness-oriented objective function, which focuses on reducing disparities in classification performance across different groups.” Can you explain the specific loss function for the discriminator? Is this just a binary classifier trained on the target attribute?
- In Section 4.1.1, *M* isn’t defined explicitly. I assume this refers to the model?
- Regarding the datasets selected -- while MNLI is a great dataset, I would not refer to it as a fairness dataset. Its usage is primarily (non-fairness based) natural language inference. I’d just reword L311: “We conduct experiments on three widely-used fairness benchmark datasets”. There are also NLP datasets specifically designed to test model fairness like Winogender, HolisticBias, CrowSPairs.
- As another more minor point, why is the sensitive attribute “blond hair” instead of “male” for CelebA?

**Comments**

- While I broadly agree that finetuning ML models to reduce bias tends to decrease performance, there are instances in NLP where model performance holds, including [1] and [2] below.
- L325: I suggest saying “African American” instead of “African”. Though HateXplain uses this label, there’s not a notion of the “African ethnicity”. It may seem like a minor point but I want to really emphasize that Africa is a huge continent with thousands of ethnicities. Additionally, it’s a separate concept from “African American” or Black, which I believe is the target group being referenced. Otherwise, the target attribute axis should not be referred to as “race”. Given the subject matter of the paper, these differences are quite critical.
- L390: Similar to the point above, I would strongly consider rewording “FairLoRA Africa” to something like “FairLoRA African American”, “FairLoRA Black American” or “FairLoRA Race”. Saying the work is to mitigate racial bias “(FairLoRA Africa)” conflates the continent of Africa with race.


**Small typos**

- L126: “can significantly reducing” → “can significantly reduce”
- L130: “may worsens fairness” → “may worsen fairness”

**References**

[1] Perturbation Augmentation for Fairer NLP

[2] Causal-Debias: Unifying Debiasing in Pretrained Language Models and Fine-tuning via Causal Invariant Learning

---

> ### Author Response · Authors · 2024-11-25
>
> ## **Response to Q1:**
>
> Thank you for your thoughtful question regarding the loss function used for training the discriminator. Below is a concise explanation:
>
> ### **Loss Function Used:**
>
> We employ the **GroupDRO loss function** to train the discriminator, as it is specifically designed to address fairness-related tasks by minimizing disparities in classification performance across groups.
>
> ### **Framework Flexibility:**
>
> FairLoRA is not limited to GroupDRO and can integrate other loss functions, such as binary cross-entropy for standard tasks or advanced fairness-oriented losses tailored to specific objectives.
>
> ### **Core Philosophy:**
>
> FairLoRA enhances fairness through post-processing fine-tuning without compromising the model's original performance. The choice of loss function impacts fairness improvement but does not degrade model performance.
>
> We have revised the manuscript to clarify the discriminator's training process and the role of the loss function. Your feedback has been invaluable in improving our work.
>
> ---
>
> ## **Response to Q2:**
>
> Thank you for pointing this out. You are correct that \(M\) refers to the model. We will explicitly define this notation in the revised manuscript to ensure clarity and avoid any potential confusion for readers.
>
> ---
>
> ## **Response to Q3:**
>
> Thank you for your insightful comment regarding the dataset selection. Below is our detailed explanation:
>
> ### **Choice of Datasets (CelebA, MultiNLI, HateXplain):**
>
> We selected CelebA, MultiNLI, and HateXplain to ensure consistency and comparability with mainstream methods, such as GroupDRO, DFR, and Lu et al., which widely use these datasets to evaluate fairness through metrics like worst-group accuracy. These datasets have become standard benchmarks for assessing fairness in classification tasks and are particularly suited for calculating metrics that our method directly targets. Using the same datasets enables a fair and transparent comparison of FairLoRA's performance with these established approaches.
>
> ### **Alternative Datasets:**
>
> We acknowledge that datasets like Winogender, HolisticBias, and CrowSPairs are also fairness-focused and designed for classification tasks. However, our primary objective was to maintain comparability with prior work and leverage widely-accepted benchmarks in fairness research that specifically focus on biases caused by imbalanced sensitive attributes. Exploring these additional datasets in future work could provide complementary insights but was beyond the scope of the current study.
>
> We hope this response addresses your concerns and clarifies the rationale behind our dataset choices.
>
> ---
>
> ## **Response to Q4:**
>
> The proportion of “male” in the CelebA dataset is 41.94%, whereas the proportion of “blond hair” is only 14.91%. The goal of our study is to mitigate bias introduced by the imbalance in sensitive attributes during training. By selecting “blond hair” as the sensitive attribute and “male” as the label, we ensure that the observed biases are due to the imbalance in the sensitive attribute, which aligns with the research objective of our work. Conversely, if we were to reverse this setup (i.e., using “male” as the sensitive attribute), the observed bias would primarily stem from the imbalance in the label rather than the sensitive attribute, which would not align with the issue we aim to address.
>
> We hope this explanation clarifies our rationale, and we appreciate your attention to this detail.

---

> ### Author Response · Authors · 2024-11-25
>
> ## **Response to W1:**
>
> Thank you for your review and comments. Regarding your concern about FairLoRA requiring balanced datasets, we provide the following clarifications:
>
> ### **FairLoRA's Objective:**
>
> FairLoRA aims to mitigate unfairness in machine learning models while maintaining or even slightly improving their original performance. A key innovation of our approach lies in the design of a discriminator, which effectively avoids the trade-off between performance and fairness—an issue that is prevalent in many existing fairness-enhancing methods.
>
> ### **Use of Balanced Datasets:**
>
> FairLoRA is a highly flexible framework that can be integrated with other fairness-enhancing techniques, such as GroupDRO, and does not rely on any specific dataset or method. In our study, we chose the simplest balanced dataset approach to demonstrate FairLoRA's capability to improve fairness without compromising performance. However, the framework can be extended to more complex settings using alternative methods.
>
> ### **Addressing Minority Samples:**
>
> FairLoRA leverages the representational capacity of pre-trained models, which allows it to remain effective even with limited samples from minority groups. By focusing on post-processing fine-tuning, FairLoRA ensures both fairness enhancement and performance preservation, even in data-scarce scenarios.
>
> We hope these clarifications address your concerns, and we have also made adjustments to the manuscript to improve its descriptions and explanations for better reader comprehension.
>
> ---
>
> ## **Response to W2:**
>
> Thank you for your valuable suggestion. We have conducted additional experiments on CelebA using another combination of attributes to further validate FairLoRA. The results, as shown in Table 5, are consistent with our previous findings, demonstrating FairLoRA's robustness and effectiveness across different attribute combinations.
>
> #### Table 5: Performance comparison across different attributes of CelebA dataset
> |               | **Heavy Makeup** |                |                | **Wearing Lipstick** |                |                |
> | ------------- | ---------------- | -------------- | -------------- | -------------------- | -------------- | -------------- |
> | **Method**    | **ACC↑ (%)**     | **WGA↑ (%)**   | **EOD↓ (%)**   | **ACC↑ (%)**         | **WGA↑ (%)**   | **EOD↓ (%)**   |
> | **ERM**       | 95.8 ± 0.1       | 45.4 ± 3.2     | 27.9 ± 1.9     | 95.8 ± 0.1           | 57.4 ± 3.5     | 29.3 ± 2.4     |
> | + FL Min.     | 95.8 ± 0.1       | **54.5 ± 3.1** | **24.4 ± 1.7** | 95.8 ± 0.2           | **63.0 ± 2.7** | **25.1 ± 2.0** |
> | **GroupDRO**  | 94.4 ± 0.5       | 65.4 ± 2.7     | 25.8 ± 1.6     | 94.4 ± 0.5           | 70.2 ± 2.5     | 25.9 ± 1.9     |
> | + FL Min.     | 94.4 ± 0.4       | **70.1 ± 2.5** | **22.7 ± 1.5** | **94.5 ± 0.4**       | **74.3 ± 2.4** | **22.5 ± 1.9** |
> | **DFR**       | 94.3 ± 1.4       | 58.0 ± 2.2     | 27.0 ± 1.8     | 94.3 ± 1.4           | 68.1 ± 1.9     | 26.7 ± 1.8     |
> | + FL Min.     | **94.5 ± 1.5**   | **63.8 ± 1.9** | **24.1 ± 2.0** | **94.4 ± 1.4**       | **73.2 ± 2.0** | **22.3 ± 1.7** |
> | **Lu et al.** | 95.4 ± 0.4       | 61.4 ± 2.5     | 28.0 ± 2.2     | 95.4 ± 0.4           | 67.8 ± 2.1     | 27.5 ± 1.7     |
> | + FL Min.     | **95.6 ± 0.5**   | **69.8 ± 2.9** | **23.2 ± 2.5** | **95.4 ± 0.4**       | **74.1 ± 2.3** | **23.1 ± 1.5** |
>
> ---
>
> ## **Response to Comments:**
>
> The methods presented in [1] and [2] focus on reducing biases in pretrained models without compromising downstream task performance. These approaches are highly effective as upstream solutions but may face limitations when addressing biases introduced during fine-tuning on specific downstream tasks. In scenarios where the fine-tuning dataset itself is imbalanced, new biases can emerge that upstream methods cannot rectify. This is the key problem our research aims to address.
>
> For instance, pretrained models fine-tuned on specific tasks such as classification in CelebA or MultiNLI often rely on datasets with inherent imbalances. In CelebA, attributes like “Blond hair” (14.91%) and “Heavy makeup” (14.37%) are underrepresented, which can lead to biases against these minority groups in the fine-tuned models. These biases may manifest as disproportionate errors for underrepresented groups, and such biases may be discovered over time as the model is deployed. In these cases, it becomes crucial to enhance fairness without re-training the model or sacrificing its original performance—a challenge our proposed method, FairLoRA, is specifically designed to address. To the best of our knowledge, FairLoRA is the first method that can effectively improve fairness in this context while preserving the original model's performance.
>
> Additionally, thank you for pointing out the terminology issue with "African" and some other grammatical errors. We have made the necessary corrections. Thank you for your thoughtful suggestions!

---

> ### Comment · Reviewer_ug8P · 2024-12-01
> **Response to Author's Rebuttal**
>
> Thank you for the detailed responses to my questions/comments!
>
> #### **Responses to Questions**
> *Q1/Q2:* That clarity makes sense, thanks!
>
> *Q3:* The desire to maintain comparison to other works is also understandable. While it doesn't affect my scoring at all, I do suggest the minor rewording, as MNLI is not explicitly a fairness dataset.
>
> #### **Response to W1**
> Thanks for the additional perspective! However, I'm still unclear if and whether FairLoRA works on an unbalanced dataset where the ratio of majority to minority samples is not 1:1. This is far more likely for real world examples. Does Section 3.4 (L243-L245) hold when the finetuning dataset is not explicitly balanced?
>
> #### **Response to W2**
> It's helpful to see these additional experiments and that the results hold across other attributes. For clarity, I understand "Heavy Makeup" and "Wearing Lipstick" to be the sensitive attributes but what are the labels here?
>
> #### **Response to Comments**
> This also makes sense to me. Thanks for considering the terminology shift. With respect to using "Blond Hair" as the sensitive attribute, it sounds like this is a small terminology difference as well. I would generally argue that the sensitive attributes refer to protected groups (e.g. gender minorities). However, this is a small and nuanced point in the grand scheme and doesn't limit the contributions of your work.
>
> Based on these responses, **I've updated my score** from 5 to 6.

---

> > ### Author Response · Authors · 2024-12-02
> >
> > Thank you once again for your thorough and thoughtful feedback on our work. We sincerely appreciate the time and effort you have taken to engage with our responses and provide further insights.
> >
> > Response to W1:
> > Yes, Section 3.4 (L243–L245) holds even when the finetuning dataset is not explicitly balanced. The purpose of using a balanced dataset was to facilitate fairness-focused finetuning; however, the fairness optimization mechanism is also effective with other forms of finetuning.
> >
> >
> > Response to W2:
> > Thank you for pointing this out. For clarity, in the experiments involving "Heavy Makeup" and "Wearing Lipstick," the labels represent **gender**. We apologize for any ambiguity in this description and will ensure this is explicitly clarified in the manuscript.
> >
> > Thank you again for your updated score and for recognizing the contributions of our work. Your feedback has been invaluable in helping us improve the quality and clarity of our manuscript.

---

### Meta-Review · Area_Chair_LmJU · 2024-12-18

**Metareview:**

This paper introduces FairLoRA, a finetuning approach which mitigates bias in machine learning models, without sacrificing model performance. This paper was reviewed by four knowledgeable referees who acknowledged that the proposed approach was a versatile solution (7Nfm), supporting intersectionality (ug8P, 9eQv), and with multiple metrics used for evaluation (dZWF).

The main concerns raised by the reviewers were:
1. The method requires a balanced dataset for finetuning (ug8P), and relies on the effectiveness of the adaptation step (7Nfm)
2. Experiments appeared limited (ug8P, 9eQv): e.g. using only a few attributes from CelebA (ug8P), no comparisons on-to-one with other methods (dZWF), missing details (dZWF), and unclear significance of the reported results (dZWF, 9eQv).
3. The theoretical contribution seemed overstated (7Nfm)
4. The novelty appeared incremental (9eQv)

During rebuttal and discussion the authors partially addressed the reviewers' concerns by clarifying the theoretical contributions, the effectiveness of the adaption step, and the requirement of balanced datasets. The authors also shared additional CelebA results considering different attributes, shared results on progressive debiasing, and argued for the novelty of their work. After discussion, some concerns remain. In particular, the authors did not show results on unbalanced datasets, although they claimed the FairLoRA framework was not limited by this. Moreover, the reviewers still hesitate about the significance of the presented results (small differences in performance), and therefore are not convinced of the effectiveness of the method to the extent claimed. The connection between theoretical analysis and proposed framework remains a concern as well. After discussing with the reviewers and careful consideration of the paper, the MR agrees with the concerns shared by the reviewers and recommends to reject. The MR encourages the authors to consider the feedback received to improve future iterations of their work, e.g. making their experimental validation stronger and working on the theoretical component of the paper.

**Additional Comments On Reviewer Discussion:**

Details can be found in the meta-review.

---

### Decision · Program_Chairs · 2025-01-22

Reject